# Cytoplasmic DAXX drives SQSTM1/p62 phase condensation to activate Nrf2-mediated stress response

Yi Yang[1], Thea L. Willis [1], Robert W. Button[1], Conor J. Strang[1], Yuhua Fu[2], Xue Wen[2], Portia R.C. Grayson [3], Tracey Evans[1], Rebecca J. Sipthorpe[1], Sheridan L. Roberts[1], Bing Hu [3], Jianke Zhang[4], Boxun Lu [2,5] & Shouqing Luo [1,5]

Autophagy cargo recognition and clearance are essential for intracellular protein quality control. SQSTM1/p62 sequesters intracellular aberrant proteins and mediates cargo delivery for their selective autophagic degradation. The formation of p62 non-membrane-bound liquid compartments is critical for its function as a cargo receptor. The regulation of p62 phase separation/condensation has yet been poorly characterised. Using an unbiased yeast two-hybrid screening and complementary approaches, we found that DAXX physically interacts with p62. Cytoplasmic DAXX promotes p62 puncta formation. We further elucidate that DAXX drives p62 liquid phase condensation by inducing p62 oligomerisation. This effect promotes p62 recruitment of Keap1 and subsequent Nrf2-mediated stress response. The present study suggests a mechanism of p62 phase condensation by a protein interaction, and indicates that DAXX regulates redox homoeostasis, providing a mechanistic insight into the prosurvival function of DAXX.

---

[1] Peninsula Medical School, Faculty of Medicine and Dentistry, Institute of Translational and Stratified Medicine, University of Plymouth, Research Way, Plymouth PL6 8BU, UK. [2] State Key Laboratory of Medical Neurobiology, School of Life Sciences, Fudan University, Shanghai 200438, China. [3] Peninsula Dental School, Faculty of Medicine and Dentistry, University of Plymouth, 16 Research Way, Plymouth PL6 8BU, UK. [4] Department of Microbiology and Immunology, Thomas Jefferson University, Philadephia, PA 19107, USA. [5] These authors jointly supervised this work: Boxun Lu, Shouqing Luo. Correspondence and requests for materials should be addressed to S.L. (email: shouqing.luo@plymouth.ac.uk)

Macroautophagy (hereafter autophagy) is a lysosome-dependent bulk degradation system that mediates clearance of aberrant intracellular components. Autophagy can selectively remove cargo materials, including misfolded protein aggregates[1–6], intracellular pathogens[7–9], certain organelles including damaged mitochondria[10–14], surplus peroxisomes[15,16] and ferritin[17,18]. During this process, cargo receptors selectively recruit the cargo and link it to autophagosome membranes by binding to ATG8/LC3 family proteins[19].

Metazoan cargo receptors, including p62/SQSTM1, NBR1, Optineurin and NDP52, recognise the cargo via their ubiquitin binding[1,2,9,20]. Among these receptors, p62 is best characterised to mediate autophagic clearance of polyubiquitinated cargos such as aggregated proteins[1,5]. p62 harbours a Phox and Bem1p (PB1) domain, an Atg8/LC3-interacting region (LIR) motif and a ubiquitin-associated (UBA) domain[19,21,22]. p62 binds ubiquitin through its C-terminal UBA domain to recruit polyubiquitinated cargos. The LIR mediates p62-Atg8/LC3 interaction[19]. Since Atg8/LC3 localise on autophagosomal membranes, p62 binding to Atg8/LC3 enables its recruited cargos to be selectively enclosed by autophagosomes. In oxidative stress conditions, p62 recruits Keap1 from the cytoplasmic Keap1-Nrf2 complex. As a result, freed Nrf2 translocates into the nucleus as a transcription factor, inducing the expression of a battery of Nrf2 target genes encoding antioxidant and anti-inflammatory enzymes[23–26]. Interestingly, p62 also senses oxidation stress to allow autophagy activation in vertebrates[27].

p62 body formation is essential for its cargo recruitment. Over the past few years, a plethora of intracellular body structures, including p62 bodies, nucleoli, cajal bodies, P granules and pro-myelocytic leukaemia (PML) bodies, have been shown as non-membrane-bound liquid compartments[28,29]. Such non-membrane structures represent open macromolecular assemblies at which biomaterials are packed and organised in a dynamic manner. Mounting evidence suggests that these non-membrane compartments are formed as liquid droplets through liquid–liquid phase separation, a process in which biomacromolecules demix from solution and form a separate liquid phase. Although hundreds of proteins are enriched in protein droplets, only a subset of proteins are believed to promote liquid droplet formation[30,31]. Condensation of intracellular droplets may drive the structures into more solid-like states[32]. It is crucial to elucidate the roles of the key proteins in droplet phase separation and transition/condensation. p62 bodies represent non-membrane structures that compartmentalise and concentrate specific sets of molecules. DAXX was identified as a death-associated protein[33], and later found to be required for cell survival during embryonic development[34]. Mutations in DAXX are associated with various cancers[35,36]. DAXX is characterised as a histone H3.3 chaperone enveloping histone H3.3 and H4 for their chromatin loading[37]. It is also known to interact with PML and co-localise with PML nuclear bodies[38].

We reveal that DAXX drives p62 liquid phase condensation by inducing p62 oligomerisation. The condensed p62 bodies efficiently recruit aberrant proteins and specific factors in the cytoplasm. The DAXX-driven p62 phase condensation promotes p62 recruitment of Keap1 and subsequent Nrf2-mediated stress response. The present study provides a mechanistic insight into p62 phase condensation, and the prosurvival function of DAXX.

## Results

**DAXX is a p62 interaction protein.** p62 body formation is required for its simultaneous interactions with ubiquitinated protein cargos and LC3[39]. Mounting evidence suggests that cellular body structures, including p62 bodies, form through the liquid–liquid phase separation mechanism[28,29,40]. Importantly, Sun et al.[29] have found that polyubiquitin chain critically promotes p62 phase separation and its cargo autophagic clearance. Interestingly, Zaffagnini et al.[41] consistently reported that p62-substrates efficient clustering requires the plyubiquitination of substrates. It is unknown if a chaperonic protein interaction is needed for p62 liquid phase formation and condensation. We performed an unbiased yeast two-hybrid (Y2H) screening with amino acids 1–300 of p62 as a bait to identify p62-interacting proteins, and found 38 potential p62 binding proteins (Supplementary Table 2). Among these binding candidates, DAXX was pulled out five times. This suggests that DAXX is a top candidate for p62 binding. We confirmed that DAXX was a positive candidate for p62 interaction by co-transforming DAXX and p62 into yeasts (Supplementary Fig. 1a). We further tested the direct p62-DAXX interaction by incubating bacterially expressed GST-p62 with 1-120aa deletion (GST-p62Δ120) and in vitro-translated DAXX, and subsequently pulling down GST proteins with glutathione beads. Supplementary Fig. 1b shows that DAXX was pulled down by GST-p62Δ120, suggesting the direct interaction between DAXX and p62. We also confirmed the endogenous p62-DAXX interaction in HeLa cells by either immunoprecipitating p62 or vice versa (Fig. 1a–b). We confirmed that the p62-DAXX interaction occurred in mouse striatum by immunoprecipitation (Fig. 1c). To identify the region of p62 that binds DAXX, we generated a series of p62 variants and found that PB1 domain (1-120aa) and 1-180aa were not required for p62 to interact with DAXX (Supplementary Fig. 1c). We further mapped 246-300aa of p62 as necessary for the p62-DAXX interaction (Supplementary Fig. 1d). Mutations in p62 have been found to associate with Paget's disease of bone (PDB)[42] and amyotrophic lateral sclerosis (ALS)[43,44]. We further examined if these mutations affect the p62-DAXX interaction. Interestingly, we found that the P392L mutation in p62 significantly reduced the p62-DAXX interaction (Supplementary Fig. 1e). This suggests that the C-terminal of p62 also contributes to the interaction. We found that 182-230aa in DAXX was required for the DAXX-p62 interaction (Supplementary Fig. 2a–b). Supplementary Fig. 2c shows the domain architectures of DAXX and p62, and that DAXX 182-230aa and p62 246-300aa are critical for their interaction.

We observed endogenous p62-DAXX co-localisation in HeLa cells (Fig. 1d), mouse striatum (Fig. 1e) and HAP1 cell (Fig. 1f). Interestingly, DAXX was recruited into p62 bodies, and p62 bodies with DAXX signals always exhibited stronger signals and bigger sizes (Fig. 1d–f), compared with those without DAXX signals. Of note, all of large p62 puncta co-localised with DAXX in the cytoplasm, while not all of DAXX puncta were positive for p62 (Fig. 1d–f). This suggests that DAXX would have other cytoplasmic roles independent of p62. Bimolecular fluorescence complementation (BiFC) has been often used to examine protein–protein interactions, given that two complementary fluorescent protein fragments will be joined by protein–protein interaction to produce complemented fluorescence, if the two proteins of interests have a physical interaction[45,46]. Indeed, the complemented fluorescence was observed, when DAXX and p62 were co-transfected into cells (Fig. 1g). The p62-DAXX co-localisation appeared dependent on their interaction. We showed that p62 interacted with 182-230aa of DAXX (see Supplementary Fig. 2a). Consistently, Fig. 1h shows that p62 interacted with full-length (FL)-DAXX (DAXX-FL) and amino-terminal (N) DAXX (DAXX-N), but not carboxyl-terminal (C) DAXX (DAXX-C). Indeed, p62 also co-localised with DAXX-FL and DAXX-N, but not DAXX-C (Fig. 1i).

**DAXX promotes the formation of p62 bodies.** Structural biology has revealed that DAXX envelopes a histone H3.3-H4 dimer, and specifically recognises histone H3.3[37]. Following our data

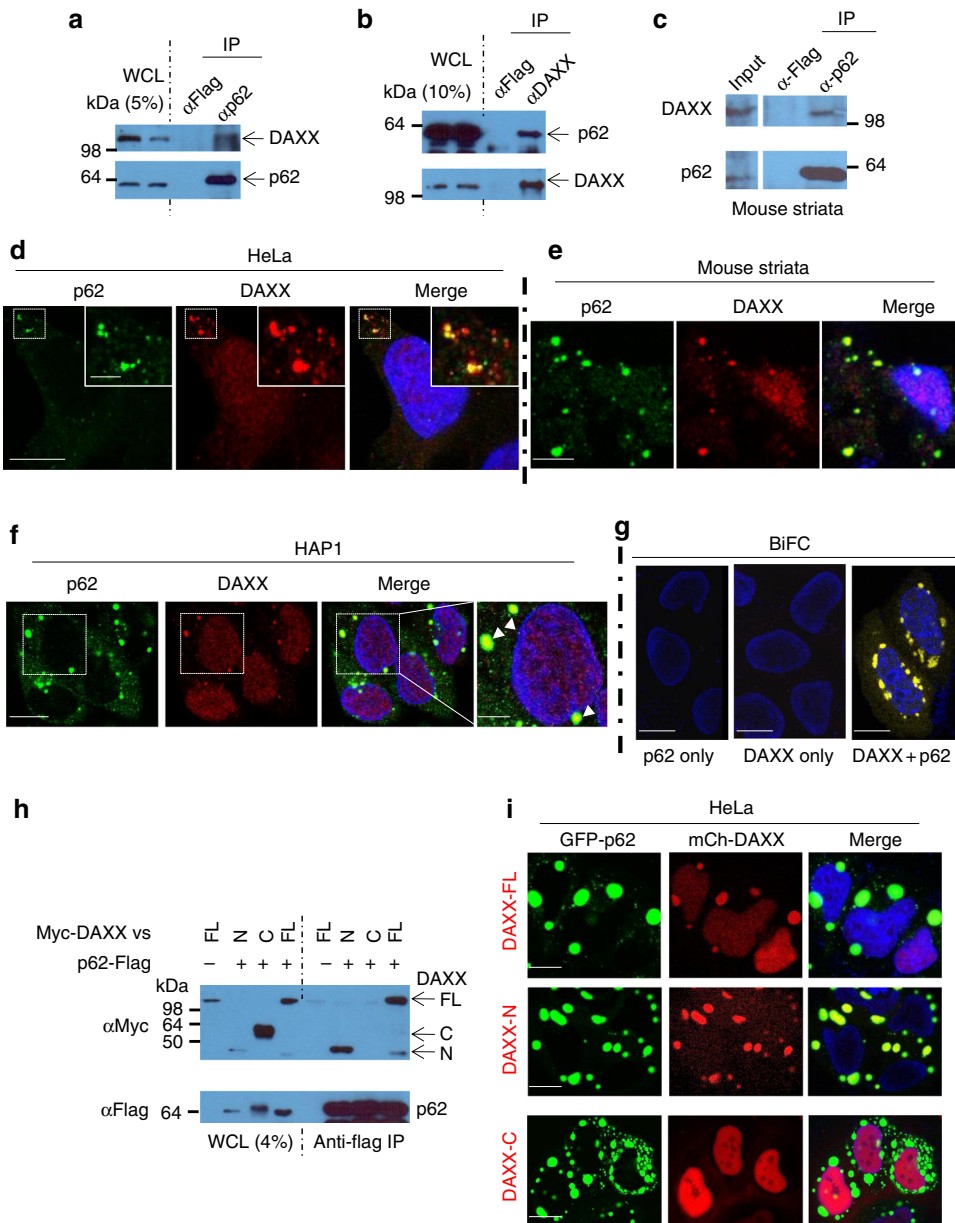

**Fig. 1** Identification of DAXX as a p62-interacting partner. **a** Endogenous p62-DAXX interaction (p62 antibody pull-down). HeLa cell lysates were subjected to Flag antibody (negative control) and anti-p62 antibody immunoprecipitation (IP). The immunoprecipitates and the whole cell lysates (WCL) were probed with anti-DAXX and anti-p62. **b** Endogenous p62-DAXX interaction (DAXX antibody pull-down). HeLa cell lysates were subjected to Flag antibody (negative control) and anti-DAXX antibody IP. The immunoprecipitates and the WCL were probed with anti-p62 and anti-DAXX. **c** Mouse (8-week) striatal lysates were immunoprecipitated by Flag antibody or anti-p62 antibody. The immunoprecipitates and the WCL were probed with anti-DAXX and anti-p62. **d** Co-localisation of endogenous p62 and DAXX in HeLa cells. HeLa cells were immunostained with anti-p62 and anti-DAXX. Images were acquired with a confocal microscope. Bar: 10 μm; bar (inset): 2 μm. **e** Mouse (6-week) striatal sections were stained with anti-p62 and DAXX antibodies. Images were acquired with confocal microscopy. Scale bar: 10 μm. **f** Co-localisation of endogenous p62 and DAXX in HAP1 cells. HAP1 cells were stained with anti-p62 and anti-DAXX. The arrowheads mark the examples of the co-localised puncta. Bar: 5 μm; bar (magnified): 2 μm. **g** p62-DAXX BiFC assay. HeLa cells were transfected with Venus BiFC plasmids expressing p62 (BiFC-VN173-p62) only, DAXX (BiFC-VC155-DAXX) only or p62 (BiFC-VN173-p62) and DAXX (BiFC-VC155-DAXX). Images were acquired with a confocal microscope at the EYFP channel. Bar: 10 μm. **h** p62 interacts with DAXX-N, but not DAXX-C. Myc-DAXX full-length (FL, 1-740aa)/vector (for negative control), Myc-DAXX FL/p62-Flag, Myc-DAXX-N (1-370aa)/p62-Flag or Myc-DAXX-C (371-740aa)/p62-Flag were co-transfected into HeLa cells. After 20 h, the cells were lysed and anti-Flag (M2) beads were used for IP. The immunoprecipitates and the WCL were probed with anti-Myc or anti-Flag antibody. vs: variants. **i** p62 co-localises with DAXX-N, but not DAXX-C. GFP-p62 was co-transfected into HeLa cells with mCherry-DAXX FL, mCherry-DAXX-N (1-370aa) or mCherry-DAXX-C (371-740aa). After 20 h, the cells were fixed, and images were acquired by a confocal microscope. The co-localisation between GFP-p62 and mCherry -DAXX variants was assessed by yellow puncta. Bar: 10 μm

indicating that DAXX and p62 have a direct interaction, we asked if the interaction has functional effects for either of the proteins. Interestingly, the sizes of GFP-p62 bodies were largely increased in the cells where DAXX was overexpressed (Supplementary Fig. 3a). We found that the size of the largest puncta in each cell, and the number of puncta over a certain size in each cell or the percentage of cells with puncta over certain size could reliably reflect the overall size extent of p62 bodies. As such, these criteria were used to quantify the size of p62 bodies in this study. Our initial data suggest that DAXX could markedly enhance the formation of p62 bodies (Supplementary Fig. 3a). We intended to exclude the possibility that the variation in transfection efficiency could lead to the difference in p62 body size. Thus, we generated Tet-on-inducible p62 stably expressing HeLa cells. In the cells stably harbouring p62-GFP, DAXX expression (along with p62-GFP expression induced by doxycycline (Dox) significantly increased p62-GFP foci/puncta formation, but markedly decreased the diffused p62-GFP (Fig. 2a). Interestingly, DAXX 178-417aa, the DAXX histone H3.3-binding domain (DX-HBD)[37] also similarly promoted p62 body formation, much like FL-DAXX (Supplementary Fig. 3b). To exclude any potential effect of DAXX on p62-GFP expression in the Tet-on cells, which could affect p62 body formation, we first induced p62-GFP expression, and then turned off its expression while DAXX was transfected into the cells. Consistently, DAXX expression also increased p62 body formation in the HeLa Tet-on cells where p62 expression was pre-induced, but turned off before DAXX was transfected (Supplementary Fig. 3c). This experiment allows us to conclude that the effect of DAXX on p62 body formation is independent of p62 expression. We also generated Tet-on inducible DAXX-expressing cell line, and consistently p62 puncta formation was increased when DAXX overexpression was induced (Supplementary Fig. 4a).

We asked if the effect of DAXX on p62 body formation was attributed to the DAXX-p62 physical interaction. Figure 2b shows that DAXX-FL and DAXX-N largely increased the sizes of p62 bodies, but DAXX-C, which did not effectively interact with p62 (see Fig. 1h), only gave minimal effects. We have shown that the P392L mutation in p62 reduced the p62-DAXX interaction (Supplementary Fig. 1e). Critically, the effect of DAXX on p62 puncta formation was also significantly reduced (Supplementary Fig. 4b). These data suggest that the physical interaction between p62 and DAXX is required for the effect of DAXX on promoting p62 body formation.

To further confirm the effect of DAXX on p62 foci formation, we generated HAP1 DAXX knockout (KO) cells using CRISPR/CAS9 system. Constantly fewer and smaller p62 foci were formed in DAXX-null HAP1 cells, in comparison with wild-type (WT) HAP1 cells (Fig. 2c). Using the CRISPR/CAS9-based lentiviral system, we also generated DAXX KO MEFs. In DAXX KO MEFs, fewer and smaller p62 puncta were formed. This was particularly notable in the presence of puromycin (Supplementary Fig. 4c). DAXX knockout causes embryonic lethality in mice[34]. We employed DAXX conditional knockout (cKO) mice to examine the in vivo role of DAXX in p62 body formation. To this end, DAXX was conditionally knocked out in T cells using lck CRE system[47]. p62 body formation was tested in DAXX-cKO T cells of the thymus with immunofluorescence. Figure 2d shows that p62 puncta in DAXX-cKO T cells in the thymus were much fewer than those in wild-type (WT) T cells of the thymus. Collectively, these data suggest that DAXX positively modulates the formation of p62 bodies.

p62 deficiency was shown to promote proteasomal activity[48,49]. We examined if DAXX could affect proteasomal activity via p62, thereby enhancing p62 puncta formation. Either DAXX knockdown (Supplementary Fig 5a) or overexpression (Supplementary Fig 5b) did not significantly alter proteasomal activity in HeLa cells. DAXX knockdown (Supplementary Fig. 5c) or overexpression (Supplementary Fig. 5d) did not significantly change p62 turnover upon the treatment of protein synthesis inhibitor cycloheximide. By qPCR, we confirmed that DAXX overexpression did not increase p62 mRNA levels (Supplementary Fig. 5e), and DAXX knockdown did not reduce p62 mRNA expression levels (Supplementary Fig. 5f). These data suggest that DAXX promotes p62 body formation not via alteration in the p62 proteasomal turnover or p62 mRNA expression.

**DAXX promotes p62 body formation independently of autophagy.** DAXX was proposed as a transcriptional repressor to negatively regulate the transcription of a set of autophagy genes and autophagy[50]. It is possible that DAXX promotes the formation of p62 bodies because it inhibits autophagy, leading to more p62 accumulation, since p62 is an established autophagy substrate[1]. To test this possibility, we first examined if DAXX promoted the formation of p62 bodies in lysosomal inhibition conditions. Supplementary Fig. 6a shows that chloroquine (CQ) treatment did not abolish the effect of DAXX on promoting p62 body formation. DAXX increased GFP-p62 body sizes in both CQ treatment and non-CQ treatment conditions.

We tested this in autophagy-competent WT mouse embryonic fibroblasts (MEFs) and autophagy-defective ATG5 KO MEFs. As shown in Supplementary Fig. 6b, the effect of DAXX on p62 sizes appeared independent of autophagy, since in both WT MEFs and ATG5 KO MEFs, DAXX significantly increased the sizes of p62 bodies. The experiments with our p62-GFP-stably expressing HeLa Tet-on cells indicate that DAXX positively modulated p62 body formation independently of variations in transfection efficiency among the cells (Fig. 2a, Supplementary Fig. 3c). We exploited this cell model to establish the irrelevance of autophagy in DAXX regulation of p62 body formation. As such, we knocked down a set of autophagy machinery genes including Atg5, Atg10 and Atg16L1. However, knockdown of these genes did not exert an effect on DAXX regulation of p62 foci formation (Supplementary Fig. 6c, Supplementary Fig. 7). Together, our data establish that DAXX increases the formation of p62 bodies independently of autophagy activity, suggesting that DAXX could promote p62 liquid phase separation/transition.

Notably, when p62-GFP HeLa Tet-on stable cells were transfected with control siRNA, Atg10 or Atg16L1 siRNA, along with vector control, overall p62-GFP exhibited bigger puncta in the cells with knockdown of Atg10 or Atg16L1, compared with those in control knockdown cells (Supplementary Fig. 6c), where the size of all p62 puncta was quantified. These suggest that p62 puncta size may be modulated by autophagy. However, the effect of autophagy on p62 puncta size was no longer observed in the cells with DAXX overexpression (Supplementary Fig. 6c). This suggests that the effect of DAXX on p62 puncta size could mask that of defects in autophagy. Interestingly, we unexpectedly observed that, in the case of Atg5 knockdown, the size of p62 puncta was always markedly reduced (Supplementary Fig. 6c). Although this phenomenon is currently not well understood, we reason that Atg5 could directly play a role in p62 puncta formation.

**DAXX drives p62 phase condensation.** Cellular body structures including p62 bodies, P granules, PML bodies and Cajal bodies are subject to phase separation[28,29,40]. Fluorescence recovery after photobleaching (FRAP) shows exchange of molecules between droplets and surrounding solution[28,29]. We confirmed that GFP-p62 fluorescence signal could undergo recovery after photobleaching in HeLa cells (Supplementary Fig. 8). The recovery rate of FRAP reflects the motile ability of droplets. Our

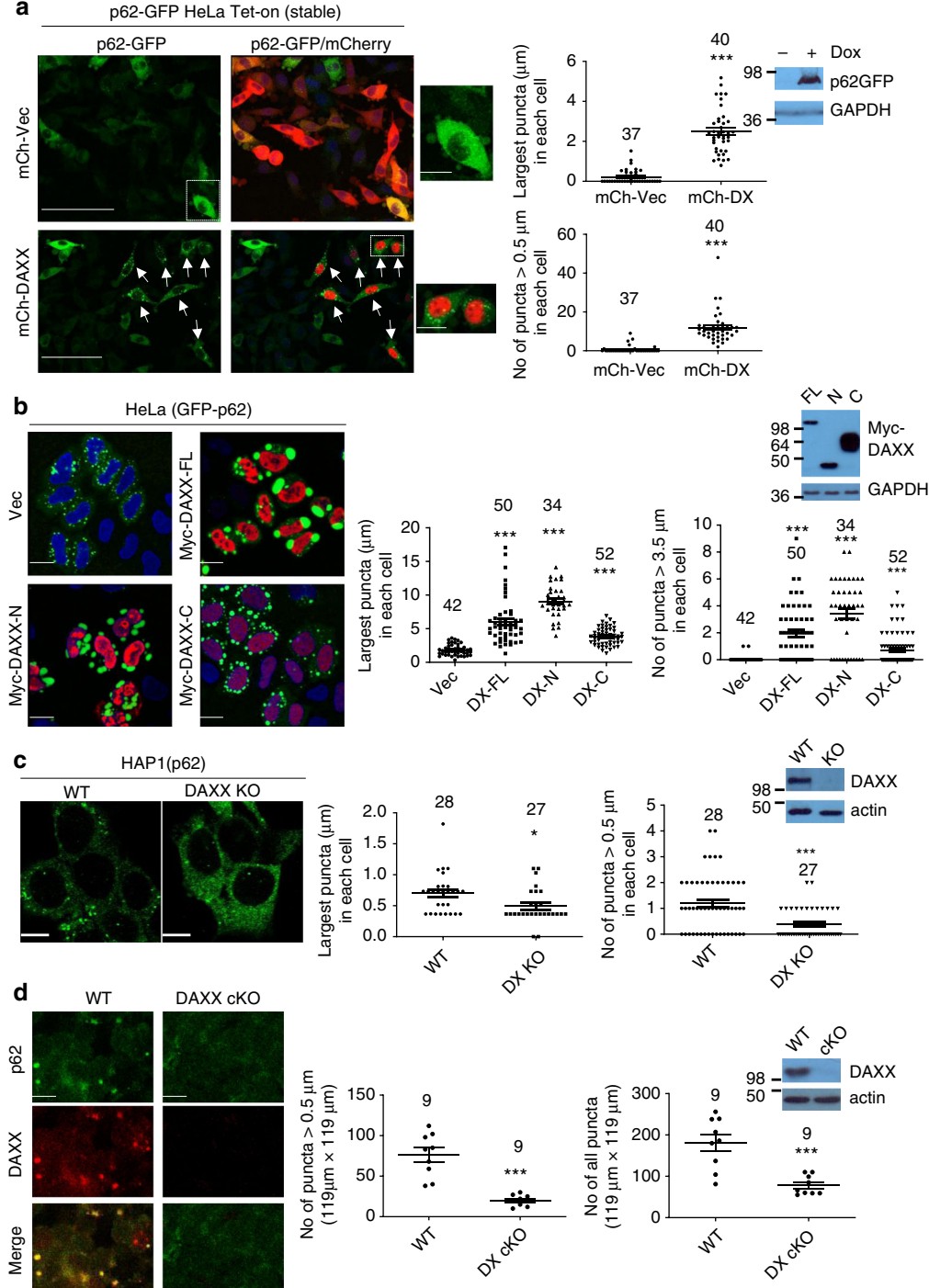

data agree that p62 bodies are viscous, as previously suggested[29]. Given that DAXX promoted p62 puncta formation, we investigated if DAXX drove p62 body phase condensation. In the presence of DAXX, the fluorescence signal of p62 recovered more slowly after photobleaching than that in the absence of DAXX (Fig. 3a). This experiment suggests that DAXX could induce p62 bodies to gel-like structures. We further examined if the effect of DAXX on p62 phase condensation was autophagy-dependent. Our data show that in both WT MEFs and ATG7 KO MEFs, DAXX significantly reduced the recovery levels after photobleaching (Fig. 3b–c), indicating that the role of DAXX in the FRAP property of p62 bodies is independent of autophagy. Proteinase K treatment is often used to examine the density of

protein aggregates/bodies because highly densed protein bodies resist proteinase K digestion. Figure 3d shows that the sizes of proteinase K-treated GFP-p62 bodies in the presence of DAXX were significantly larger than those in the absence of DAXX. Oligomerised/aggregated proteins could be soluble in chaotropic agents such as 8 M urea-containing buffers[51]. To consolidate our observation that DAXX promotes p62 phase condensation, we examined the solubility of DAXX-induced p62 oligomers in the Nonidet P-40 (NP-40) or urea-containing buffer. Our data show that a vast majority of DAXX-induced p62 oligomers were NP-40 insoluble, but 8 M urea-soluble (Fig. 3e). Together, these data suggest that DAXX drives p62 phase condensation.

**Fig. 2** DAXX promotes p62 body formation. **a** HeLa Tet-on cells stably harbouring p62-GFP were transfected with mCherry-vector or mCherry-DAXX. Three hours after the transfection, p62-GFP expression was induced for 20 h. Confocal images were acquired. Bar: 50 μm; bar (magnified): 10 μm. The arrows mark the cells with p62 bodies. The diameter of the biggest p62-GFP puncta in each cell was measured (LAS). The number of p62-GFP puncta > 0.5 μm in each cell was assessed (ImageJ). n = 37 (mCh-Vec), 40 cells (mCh-DAXX) from three independently plated wells. Statistical analysis was performed with unpaired/two-tailed T-tests. Data are shown as mean ± sem. ***P < 0.0001. Immunoblot shows p62-GFP expression. **b** GFP-p62 was co-transfected into HeLa cells with vector, Myc-DAXX, Myc-DAXX-N (1-370aa) or Myc-DAXX-C (371-740aa). After 20 h, the cells were stained with anti-Myc. Confocal images were acquired. Bar: 20 μm. The diameter of the biggest GFP-p62 puncta in each cell was measured (LAS). The number of GFP-p62 puncta > 3.5 μm in each cell was assessed (LAS). n = 42 (Vec), 50 (Myc-DAXX-FL), 34 (Myc-DAXX-N), 52 cells (Myc-DAXX-C) from three independently plated wells. Statistical analysis was performed by one-way ANOVA (Tukey's test). Data are shown as mean ± sem. ***P < 0.0001. Immunoblot shows DAXX expression. **c** HAP1 WT and DAXX KO cells were stained with anti-p62. Confocal images were acquired. Bar: 10 μm. The diameter of the biggest p62 puncta in each cell was measured (LAS). The number of p62 puncta > 0.5 μm in each cell was assessed (LAS). n = 28 (WT), 27 cells (KO) from three independently plated wells. Statistical analysis was performed with unpaired/two-tailed T-tests. Data are shown as mean ± sem. *P = 0.0136; ***P < 0.0001. Immunoblot shows DAXX expression. **d** The thymus from WT and DAXX conditional KO mice were subjected to immunofluorescence for p62/DAXX. Confocal images were acquired. Bar: 10 μm. The number of p62 puncta > 0.5 μm in each image frame (119 μm × 119 μm) was assessed (ImageJ). The number of all p62 puncta in each image frame was assessed (ImageJ). n = 9 biologically independent samples for each group. Statistical analysis was performed with unpaired/two-tailed T-tests. Data are shown as mean ± sem. ***P < 0.0001 (left); ***P = 0.0003 (right). Immunoblot shows DAXX expression

**DAXX promotes p62 phase separation in vitro**. To examine if DAXX directly promotes p62 phase separation, we purified recombinant p62 and DAXX expressed in *E. coli* (Fig. 4a). Differential interference contrast (DIC) microscopy showed that recombinant p62 only formed few droplet-like shapes at a concentration of 5 μM in phase separation buffer (Fig. 4b). These data are in agreement with previous reports showing that p62 alone only has relatively low basal activity in phase separation[29,41]. Five micromolar volume of DAXX markedly increased p62 droplet formation. In the presence of DAXX, both size and number of p62 droplets were enhanced (Fig. 4b). We confirmed that buffer only or DAXX alone did not form droplets (Fig. 4b). The phase-separated liquid droplets are expected to have a sphere-like shape. When liquid-like droplets undergo transitions to form viscous/gel-like assemblies, they can have deformed spherical or irregular structures[52]. p62 droplets/ assemblies have been suggested to be viscous structures[29,41]. Our evidence also suggests that p62 droplets, particularly with DAXX, exhibit gel-like properties (Fig. 3a–c). Therefore, we expect that p62 assemblies could have deformed spherical structures.

The levels of p62 droplet formation correlated with the dose of DAXX added (Fig. 4c). This indicates that DAXX is critical to drive p62 droplet formation. These droplets exhibited fusion characteristic (Fig. 4d). This indicates these droplets have liquid-like properties. We separated droplets from the phase separation solution by sedimentation and determined the levels of p62 and DAXX in each fraction by immuoblotting. Figure 4e confirms that the mixture with DAXX yielded more p62 protein in the pellets than the mixture containing no DAXX. The in vitro reconstituted system firmly suggests that DAXX is critical for p62 phase separation.

**DAXX promotes p62 oligomerisation**. Since DAXX promotes p62 phase condensation independently of p62 transcription and autophagy, we postulate that DAXX may increase p62 oligomerisation. To test this hypothesis, we examined the interaction affinity between p62 molecules in the absence or the presence of DAXX. Thus, we examined the interaction between p62-HA and p62-Flag. Figure 5a shows that more p62-HA was pulled down by p62-Flag in the presence of DAXX. These suggest that DAXX may increase p62-p62 interaction affinity, and thus promote p62 oligomerisation. We investigated the role of DAXX in p62 oligomerisation by using native gel electrophoresis, which is a standard approach for protein oligomerisation assays[53]. In HeLa cells expressing GFP-p62 along with DAXX, GFP-p62 was prone to forming more highly oligomerised p62 (Fig. 5b). Interestingly, we observed more GFP-p62 breakdown products in the cells with

both GFP-p62 and DAXX (Fig. 5b). This indicates that oligomeric p62 tends to be broken down by autophagy. Komatsu and colleagues suggest that the levels of free GFP positively correlate with autophagic degradation rate of GFP-p62 in cells[54]. Our data confirm that the levels of free GFP were consistent with GFP-p62 autophagic degradation (Supplementary Fig. 9a).

We also examined if SDS-PAGE could distinguish the levels of p62 oligomerisation in the cells expressing p62 alone or p62 along with DAXX. To this end, p62-HA was transfected into HeLa cells with either control vector or Flag-DAXX. In the presence of Flag-DAXX, more oligomeric p62 was present in SDS-PAGE conditions (Fig. 5c). We used WT HAP1 and DAXX KO HAP1 cells to test endogenous p62 oligomerisation. Using the Wes size assay (a quantitative capillary electrophoresis), we confirmed that, in native conditions, much less oligomeric p62 was formed in DAXX KO HAP1 cells, compared with that in WT HAP1 cells (Fig. 5d). We also observed p62 oligomerisation in WT HAP1 cells, but not in DAXX KO HAP1 cells in SDS-PAGE conditions (Fig. 5e). Moreover, DAXX-induced p62 oligomerisation in both WT and ATG5 KO MEFs (Fig. 5f), confirming that the effect of DAXX on p62 oligomerisation is independent of autophagy. Our in vitro assay shows that DAXX 1-250aa was sufficient to induce p62 oligomerisation/dimerisation (Fig. 5g). Taken together, these data establish that DAXX induces p62 oligomerisation that is predicted a key step for p62 phase condensation/separation.

**DAXX determines p62 oligomerisation independently of PB1**. It is known that the PB1 domain (1-120aa) is critical for p62 oligomerisation[21]. Since the region of p62 binding DAXX locates at 246-300aa, to further establish that DAXX promotes p62 oligomerisation, we employed GFP-p62 with PB1 deletion (GFP-p62ΔPB1), which is incompetent to oligomerise in normal conditions, and tested that if DAXX could promote the oligomerisation of GFP-p62ΔPB1. Figure 6a shows that DAXX markedly increased the formation of GFP-p62ΔPB1 foci-like structures. The effect of DAXX on GFP-p62ΔPB1 oligomerisation appeared to be more pronounced in the cells undergoing the treatment of puromycin, a proteotoxin that causes protein misfolding[55]. To exclude the potential role of endogenous p62 in DAXX-induced PB1Δ-p62 puncta formation, we examined the effect of DAXX on the puncta formation of p62 K7A D69A (the PB1-inactivated p62 mutant) in p62 KO MEFs. Supplementary Fig. 9b shows that DAXX significantly induced the puncta formation of p62 K7A D69A and WT-p62 in p62 KO MEFs. Using native gel electrophoresis, we confirmed that DAXX-induced GFP-p62ΔPB1

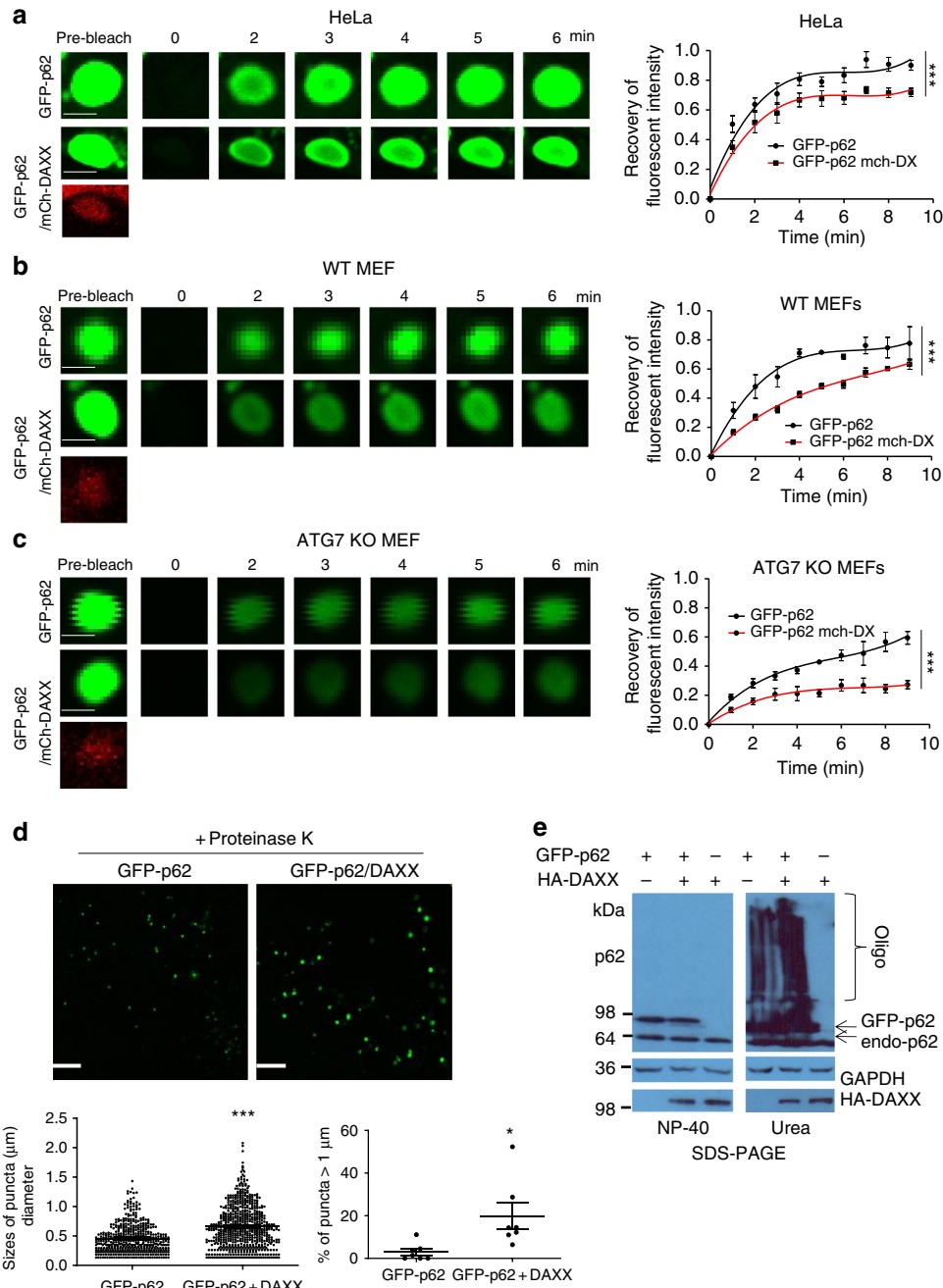

**Fig. 3** DAXX induces p62 liquid phase condensation. **a**–**c** HeLa cells (**a**), wild-type (WT) mouse embryonic fibroblasts (MEFs) (**b**) or ATG7 KO MEFs (**c**) were transfected with GFP-p62 or GFP-p62/mCherry-DAXX. After 24 h, cells were subjected to live imaging and FRAP assays. $n = 6$ for each group (**a**), 7 for each group (**b**), 8 for each group (**c**) puncta from three independently plated wells. The data were plotted and analysed with Graphpad Prism V5 using nonlinear regression (curve fit). Two-way ANOVA was used for statistical analysis. Data are shown as mean ± sem. ***$P < 0.0001$ (all). Bar: 2 μm. **d** HeLa cells were transfected with GFP-p62/vecor or GFP-p62/Flag-DAXX (1:2). After 20 h, the cells were treated with proteinase K (50 μg/ml) at 37 °C for 30 min. Images were acquired with a fluorescence microscope. $n = 475$ (GFP-p62), 628 (GFP-p62 + DAXX) puncta from three independently plated wells (first); $n = 7$ independently plated samples for each group (second). Statistical analysis was performed with unpaired two-tailed T-tests. Data are shown as mean ± sem. ***$P < 0.0001$. *$P = 0.0177$. **e** Oligomerised p62 induced by DAXX is urea-soluble. GFP-p62/vector, GFP-p62/HA-DAXX or HA-DAXX/vector were transfected into HeLa cells. After 20 h, the cells were extracted in Buffer A with NP-40. The supernatants, and the pellets dissolved with Buffer A containing 8 M urea, were subjected to SDS-PAGE, and probed with anti-p62, and other antibodies as indicated. Oligo: oligomers

oligomerisation (Fig. 6b). In SDS-PAGE conditions, p62ΔPB1 higher-order forms were only seen in the cells overexpressing DAXX (Fig. 6c). Importantly, our in vitro native Wes assay shows that DAXX 1-250aa was sufficient to induce p62ΔPB1 oligomerisation (Fig. 6d), confirming that DAXX promoted p62 oligomerisation independently of its PB1 domain. Therefore, we

conclude that DAXX would be a factor to determine the degree of p62 oligomerisation.

**DAXX is critical for p62 to recruit ubiquitinated proteins.** Since p62 oligomerisation is important for its recruitment of

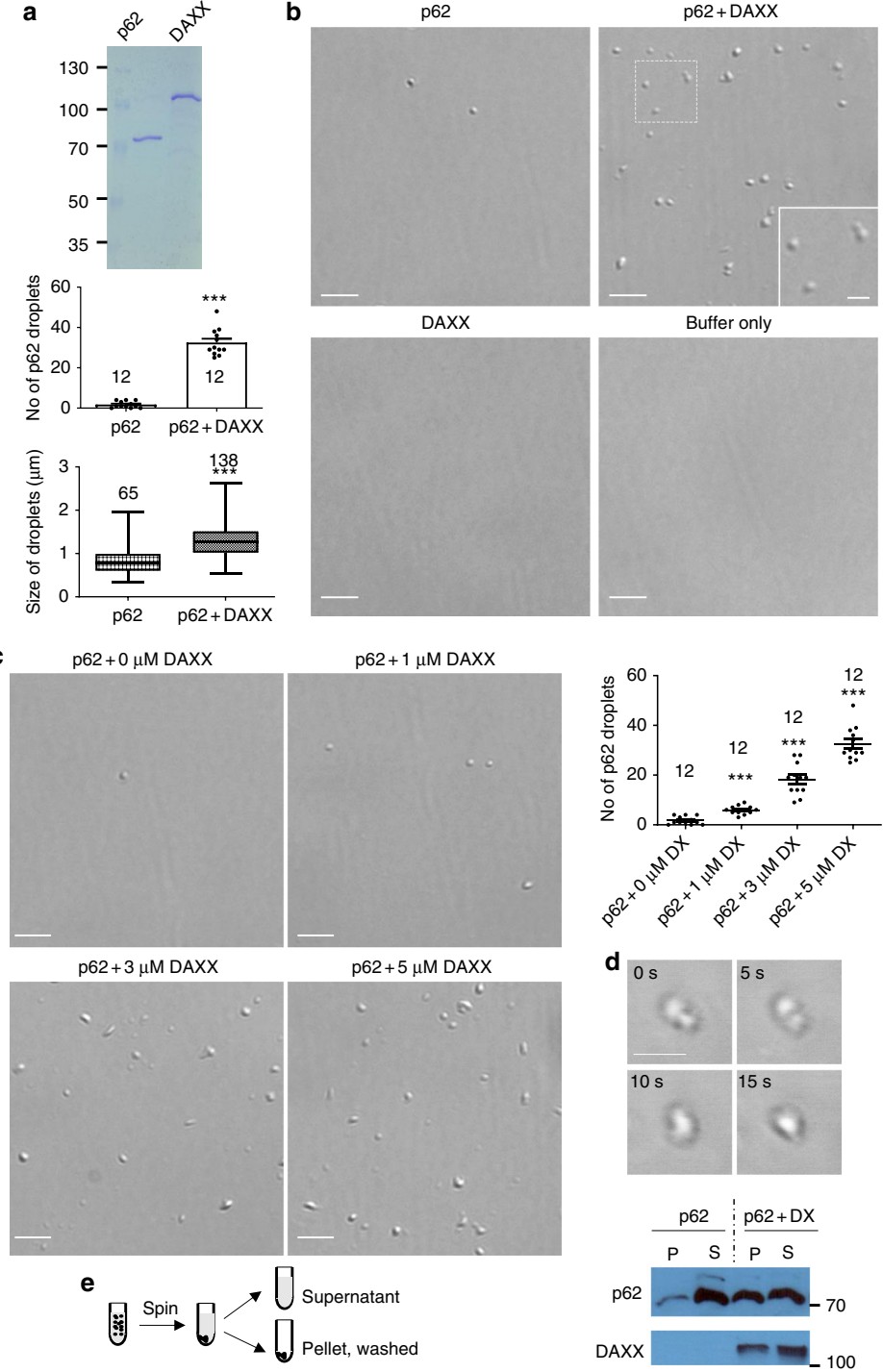

ubiquitinated proteins[39,56], we investigated if DAXX modulated the p62-polyubiqitinated proteins interaction. DAXX over-expression markedly increased p62-polyubiquitinated protein co-localisation (Fig. 7a), as well as the sizes of p62 bodies and polyubiquitinated protein foci (Supplementary Fig. 10a). Indeed, the sizes of ubiquitin-positive foci were significantly lower in DAXX KO HAP1 cells than those in WT HAP1 cells (Supplementary Fig. 10b). Similarly, in DAXX KO MEFs, the sizes of p62 and ubiquitin puncta (Supplementary Fig. 10c), and the p62-ubiquitin foci co-localisation (Fig. 7b) were much weakened, compared with these in WT MEFs. We validated the role of DAXX in the formation of ubiquitin puncta in vivo in flies. We have obtained the *Drosophila* lines with WT background

(W1118) or *UAS*-DLP (DAXX-like protein in *Drosophila*) shRNA (62353) (Bloomington). These lines were crossed with the *Elav*-GAL4 driver line to induce DLP knockdown or control in the nervous system, given that ELAV is known to express in neurons and neural progenitor cells[57]. DLP is predominantly expressed in the neurons of fly brains[58]. We performed qPCR to confirm the knockdown effect of DLP shRNA in the nervous system, with mRNAs isolated from fly brains. Supplementary Fig. 10d shows that brain DLP expression was reduced in the DLP-KD line, indicating that nervous DLP expression was knocked down in our flies. The brains of the flies with DLP-shRNA expression were dissected and stained with ubiquitin antibody. Figure 7c shows that in DLP knockdown flies, ubiquitin formed fewer puncta.

**Fig. 4** DAXX promotes p62 liquid phase separation in vitro. **a** The purified bacteria-expressed p62 and DAXX. **b** The mixture of 5 μl of 10 μM p62 and 10 μM DAXX or the indicated single protein mixed with the control buffer (5 μM final concentration) was subjected to in vitro phase separation in a microcentrifuge tube for 1 h, and imaging was carried out on a glass slide. Protein solutions were centrifuged to clear potential aggregates, immediately prior to in vitro phase separation assays. The number of p62 droplets in each image (40 μm × 40 μm) was scored (LAS-X). $n = 12$ images for each group. p62 droplet size was assessed (LAS-X). $n = 65$ (p62) or 138 (p62 + DAXX) droplets. Statistical analysis was performed with unpaired/two-tailed T-tests. Data are shown as mean ± sem. ***$P < 0.0001$. The DIC images were acquired with Leica DMi8 microscopy. Bar: 5 μm; bar (inset): 2 μm. Centre line: the median; whiskers: the minimum to maximum; box: the first to the third quartile. **c** The experimental conditions are similar as those in (**b**), except that different concentrations of DAXX protein were used. The DIC images were acquired with Leica DMi8 microscopy. Scale bar: 5 μm. p62 droplet number was quantified in each image (40 μm × 40 μm). $n = 12$ images for each group. Statistical analysis was performed by one-way ANOVA (Tukey's test). Data are shown as mean ± sem. ***$P < 0.0001$. **d** In vitro phase separation was carrried out on a well of a 384-well plate for 30 min (RT). The time-lapse DIC images were acquired with confocal microscopy. Scale bar: 4 μm. **e** The mixture (30 μl) of 15 μl of 10 μM p62 and 10 μM DAXX, or the p62 protein alone with control buffer was subjected to in vitro phase separation at RT for 1 h. After 15-min centrifugation at 16,000 × $g$, the pellet and supernatant were separated. The pellet was washed once with the phase separation buffer, and resuspended in 30 μl phase separation buffer. The diagram illustrates the assay procedures. The pellet (P) and supernatant (S) were subjected to immunoblot with indicated antibodies

We then directly detected the interaction between p62 and ubiquitinated proteins in these cells. Markedly less ubiquitinated proteins were pulled down by p62 in DAXX KO HAP1 cells (Fig. 7d), confirming that the interaction between p62 and ubiquitinated cargos in DAXX KO HAP1 cells was weaker than that in WT HAP1 cells. Consistently, in DAXX knockdown HeLa cells, less ubiquitinated proteins were pulled down by p62 immunoprecipitation in either control or puromycin treatment conditions (Fig. 7e), consolidating that DAXX is critical for p62 to recruit ubiquitinated protein cargos. Together, these data suggest that DAXX positively regulates p62-mediated ubiquitinated protein cargo recruitment due to its role in promoting p62 phase separation.

**Cytoplasmic DAXX drives Nrf2-mediated stress response**. After establishing that DAXX-induced p62 phase condensation, we aimed to examine the functional consequence of this action. Given that p62 oligomerisation reduces ROS production by recruiting Keap1 and subsequently activating the transcription factor Nrf2[23,59], we sought to test if DAXX plays a role in redox homoeostasis. To this end, we examined the co-localisation of DAXX-p62-Keap1. Our data show that the three proteins co-localised well when they were overexpressed (Supplementary Fig. 11a) or endogenously expressed (Supplementary Fig. 11b). We observed that DAXX siRNA ablation markedly reduced p62 condensation and p62-Keap1 co-localisation (Fig. 8a). These data suggest that DAXX promotes Keap1-p62 co-localisation in puncta. We confirmed that DAXX knockdown increased Keap1 levels by Western blot (Supplementary Fig. 11c), presumably due to reduced p62 recruitment and autophagic clearance in the conditions. Indeed, Nrf2 levels in nuclei were reduced when DAXX or p62 was knocked down (Fig. 8b).

The transcription of a battery of antioxidant and detoxifying genes, including NAD(P)H dehydrogenase quinone 1 (nqo1) and glutathione S-transferase Mu1 (gstm1), is regulated by Nrf2[60]. DAXX knockdown caused a reduction in the protein levels of NQO1 and GSTM1 (Fig. 8c). Consistently, Nrf2 levels in nuclei were also reduced in DAXX KO HAP1 cells (Supplementary Fig. 11d), and DAXX knockout decreased both mRNA levels and protein levels of NQO1 and GSTM1 in HAP1 cells (Supplementary Fig. 11e–f). Consequently, DAXX or p62 knockdown increased the levels of reactive oxygen species (ROS) in either the absence (Fig. 8d) or the presence (Fig. 8e) of $H_2O_2$. Dual knockdown of DAXX and p62 did not cause a further increase in ROS levels (Fig. 8d–e, Supplementary Fig. 12), suggesting that the effect of DAXX was dependent on p62. These data suggest that DAXX is a critical factor for Nrf2-mediated detoxification via driving p62 phase separation/condensation.

We reasoned that the effect of DAXX on p62 recruitment of Keap1 would result in Nrf2 subsequent activation. To assess the role of DAXX in Nrf2 activity, we performed Nrf2 recognition motif/antioxidant responsive element (ARE) luciferase reporter assays. Our data suggest DAXX or p62 knockdown caused a consistent, albeit mild, reduction in Nrf2 activity in basal conditions (Fig. 8f). However, in the cells treated with $H_2O_2$, DAXX or p62 knockdown caused more pronounced reduction in Nrf2 activity (Fig. 8g). Dual knockdown of DAXX and p62 did not cause a further reduction in Nrf2 activity (Fig. 8f–g), suggesting that the effect of DAXX in Nrf2 activity was dependent on p62. DAXX mutations associate with a variety of tumours[36]. We found that D349A mutation in DAXX significantly reduced the DAXX-p62 interaction (Supplementary Fig. 11g). WT DAXX increased Nrf2 transcriptional activity, however, the effect of DAXX D349A mutant on Nrf2 transcription was significantly lowered, consistent with the reduced interaction between the DAXX mutant and p62 (Supplementary Fig. 11h). We sought to test if DAXX deficiency sensitises cells undergoing oxidative insults. We chose HAP1 cells for the assays since these cells are more sensitive to oxidative stress. Like p62 KO HAP1 cells, DAXX KO sensitized the cells to $H_2O_2$ oxidation (Fig. 8h). Our data established that DAXX promotes p62 phase condensation, and further modulates ROS production by the Keap1-Nrf2 pathway.

**Discussion**

p62 is a multi-functional protein, playing critical roles in selective autophagy and functioning as a signalling hub for diverse cellular events, such as cell survival, amino acid sensing and the oxidative stress response[23,61–63]. Recently Carroll et al. reported that in vertebrates, p62 senses cellular redox state to allow autophagy activation in response to oxidation stress[27]. Here we characterise that DAXX facilitates p62 liquid phase separation and condensation. We initially found that DAXX promotes p62 body size. For consistency, we employed the size of p62 puncta as an approximate surrogate of p62 body formation, as used previously[29]. The ratio of punctate volume versus total cell volume would be an advantage to quantify the levels of p62 clustering. In our experimental settings, it was technically challenging to accurately quantify cellular volumes, given the complexity in the shapes of the adherent cells used in this study.

Mounting evidence suggests that non-membrane-bound liquid compartments, including nucleoli, cajal bodies, PML bodies and P granules, form through liquid–liquid phase separation in the nucleus or the cytoplasm, a process in which biomacromolecules de-mix or separate from solution and form a separate liquid phase[28,64,65]. Like membrane-bound vesicles, non-membrane-bound assemblies are essential in sorting intracellular

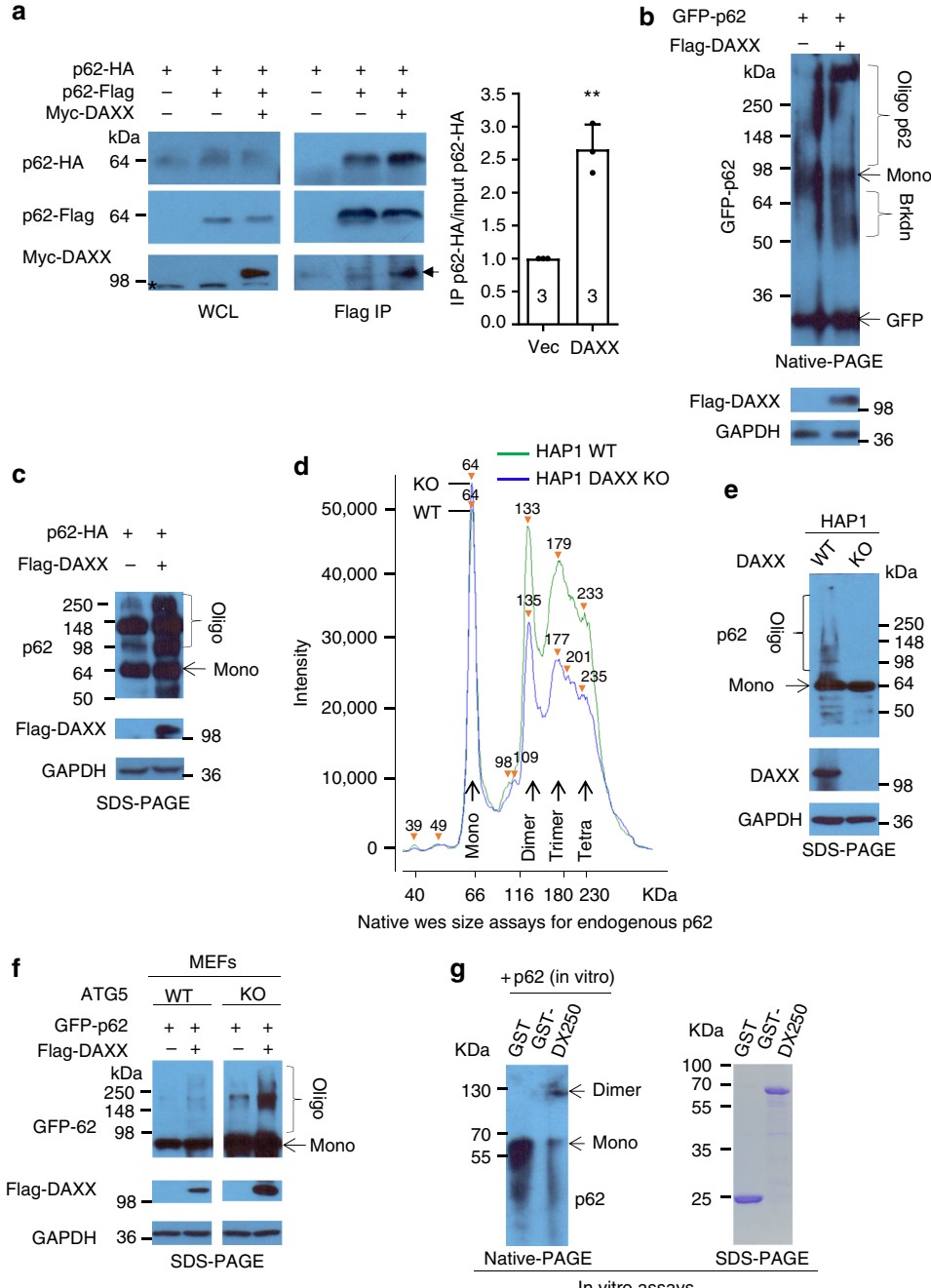

**Fig. 5** DAXX promotes p62 oligomerisation. **a** p62-HA/vector, p62-HA/p62-Flag/vector or p62-HA/p62-Flag/Myc-DAXX were transfected into HeLa cells. After 20 h, the cells were lysed, and the cell lysates were immunoprecipitated with anti-Flag (M2) agarose beads. The immunoprecipitates and the whole cell lysates (WCL) were probed with anti-HA and anti-Flag. * denotes non-specific bands. Arrow marks Myc-DAXX. The ratios of immunoprecipitated p62-HA and input p62-HA were computed (the ratios in controls are set as 1). Statistical analysis was performed with unpaired two-tailed T-tests. Data are shown as mean ± sd. $n = 3$ independent experiments. **$P = 0.0083$. **b** GFP-p62/vector or GFP-p62/Flag-DAXX were transfected into HeLa cells. After 20 h, the cells were lysed. The cell lysates were subjected to native PAGE, and probed with GFP antibody. The cell lysates were also resolved by SDS-PAGE and probed with anti-Flag and anti-GAPDH. Brkdn: breakdown products; Oligo: oligomers; Mono: monomers. **c** p62-HA/vector or p62-HA/Flag-DAXX were transfected into HeLa cells. The cell lysates were subjected to SDS-PAGE, and subsequently probed with anti-p62, anti-Flag and anti-GAPDH. **d** WT HAP1 or DAXX KO HAP1 cells were lysed, and the cell lysates were subjected to native Wes size assays with anti-p62 antibody (1:100 dilution) in native conditions (see Methods section for more details). **e** HAP1 WT or HAP1 DAXX KO cells were lysed, and the cell lysates were subjected to SDS-PAGE and probed with anti-p62, anti-DAXX and anti-GAPDH. **f** GFP-p62/vector or GFP-p62/Flag-DAXX were transfected into Atg5 WT or KO MEFs. The cells were lysed and the cell lysates were subjected to SDS-PAGE, and probed with anti-GFP, anti-Flag and anti-GAPDH. **g** In vitro-translated p62 was incubated with recombinant GST or GST-DAXX 1-250aa (see Methods). The incubated mixtures were subjected to native electrophoresis (native PAGE), and subsequently probed with p62 antibody. GST and GST-DAXX 1-250aa were subjected to SDS-PAGE

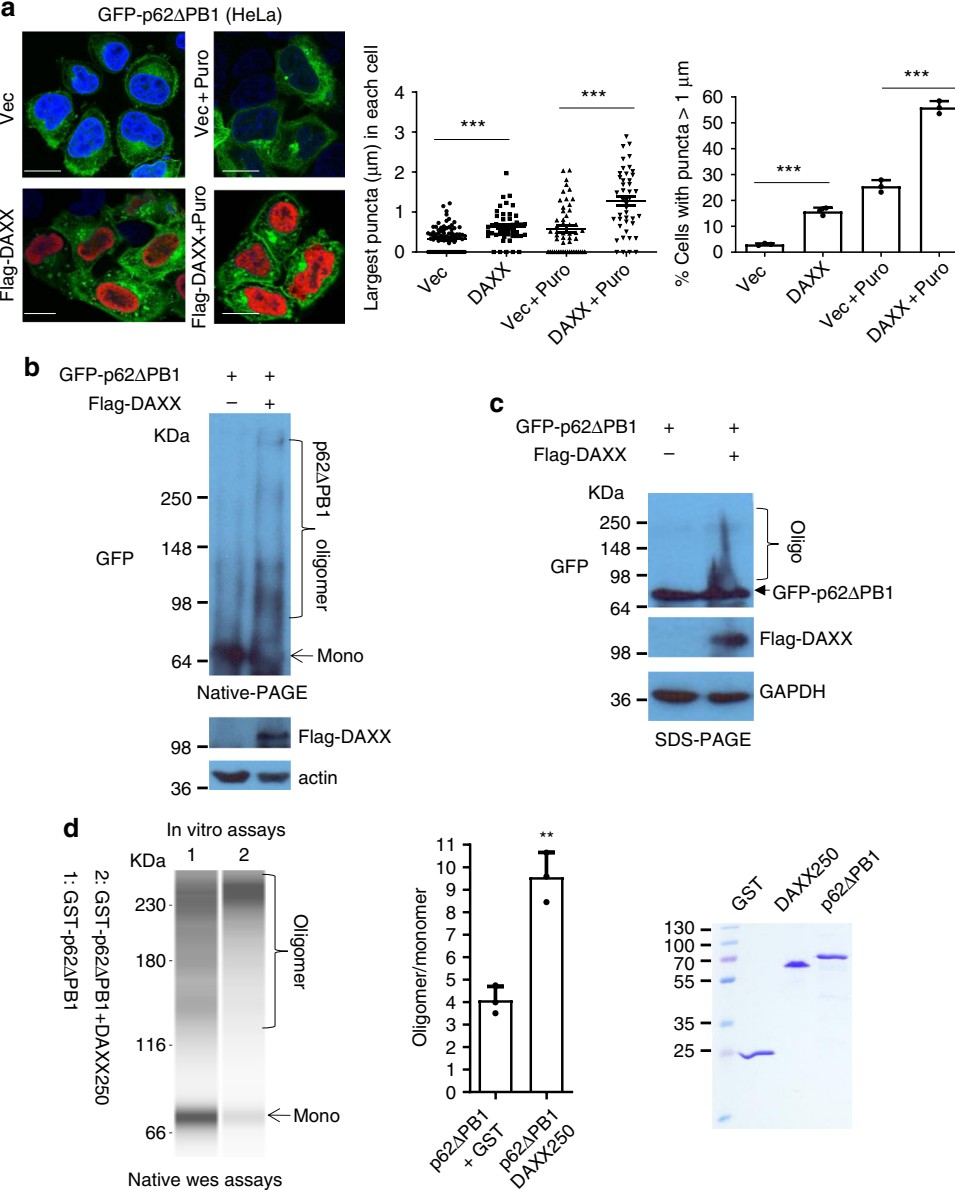

**Fig. 6** DAXX promotes p62 oligomerisation independently of PB1 domain. **a** GFP-p62 with 1-120aa deletion (GFP-p62ΔPB1) was co-transfected into HeLa cells with vector or Flag-DAXX. After 20 h, the cells were treated without or with puromycin for 5 h. The cells were fixed, and stained with anti-Flag antibody. The images were acquired with a confocal microscope. Bar: 10 μm. The diameter of the biggest GFP-p62ΔPB1 puncta in each cell was measured (LAS). $n = 88$ (Vec), 49 (DAXX), 51 (Vec + Puro), 43 cells (DAXX + Puro) from three independently plated wells. The percentage of cells with GFP-p62ΔPB1 puncta >1 μm was assessed. $n = 3$ independently plated samples for each group. Statistical analysis was performed with unpaired two-tailed T-tests. Data are shown as mean ± sem. ***$P < 0.0001$ (left); ***$P = 0.0001$ (right). **b** GFP-p62ΔPB1 was co-transfected into HeLa cells with empty vector or Flag-DAXX. After 20 h, the cell lysates were subjected to native PAGE, and immunoblot with anti-GFP, or SDS-PAGE and immunoblot with anti-Flag and anti-actin. **c** GFP-p62ΔPB1 was co-transfected into HeLa cells with empty vector or Flag-DAXX. After 20 h, the cell lysates were subjected to SDS-PAGE and immunoblot with anti-GFP, anti-Flag and anti-GAPDH. **d** Simple Wes assays. Bacterially expressed GST-p62ΔPB1 (0.5 μg) was incubated with GST or GST-DAXX 1-250aa (DAXX250) (1.5 μg) in 1× PBS at room temperature. After 48 h, the mixtures were subjected to native Wes assays with anti-p62 (1:100) (see Methods for details). The ratios of oligomeric versus monomeric GST-p62ΔPB1 were digitally recorded. $n = 3$ independent experiments. Statistical analysis was performed with unpaired two-tailed T-tests. Data are shown as mean ± sd. **$P = 0.0017$. The recombinant proteins were subjected to SDS-PAGE

biomaterials. Sun et al.[29] reported the role of polyubiquitin chain in p62 phase separation. Zaffagnini et al.[41] consistently reported that p62 cluster formation is triggered by multiple ubiquitin chains in polyubiquitinated proteins. p62 droplets may exhibit an intermediate feature between solid and liquid phases (gel-like phase)[29,41]. Interestingly, the induction of polyubiquitin chains in p62 clustering requires the PB1 domain, providing a filamentous scaffold that is structurally connected to the UBA domain[41]. By contrast, the role of DAXX in p62 phase separation/condensation is independent of the PB1. DAXX confers p62 condensates more solid-like characters by increasing its condensation. Weak and transient interactions between molecules with multivalent domains or intrinsically disordered regions are a driving force for liquid phase separation[66]. Intracellular liquid phase condensates

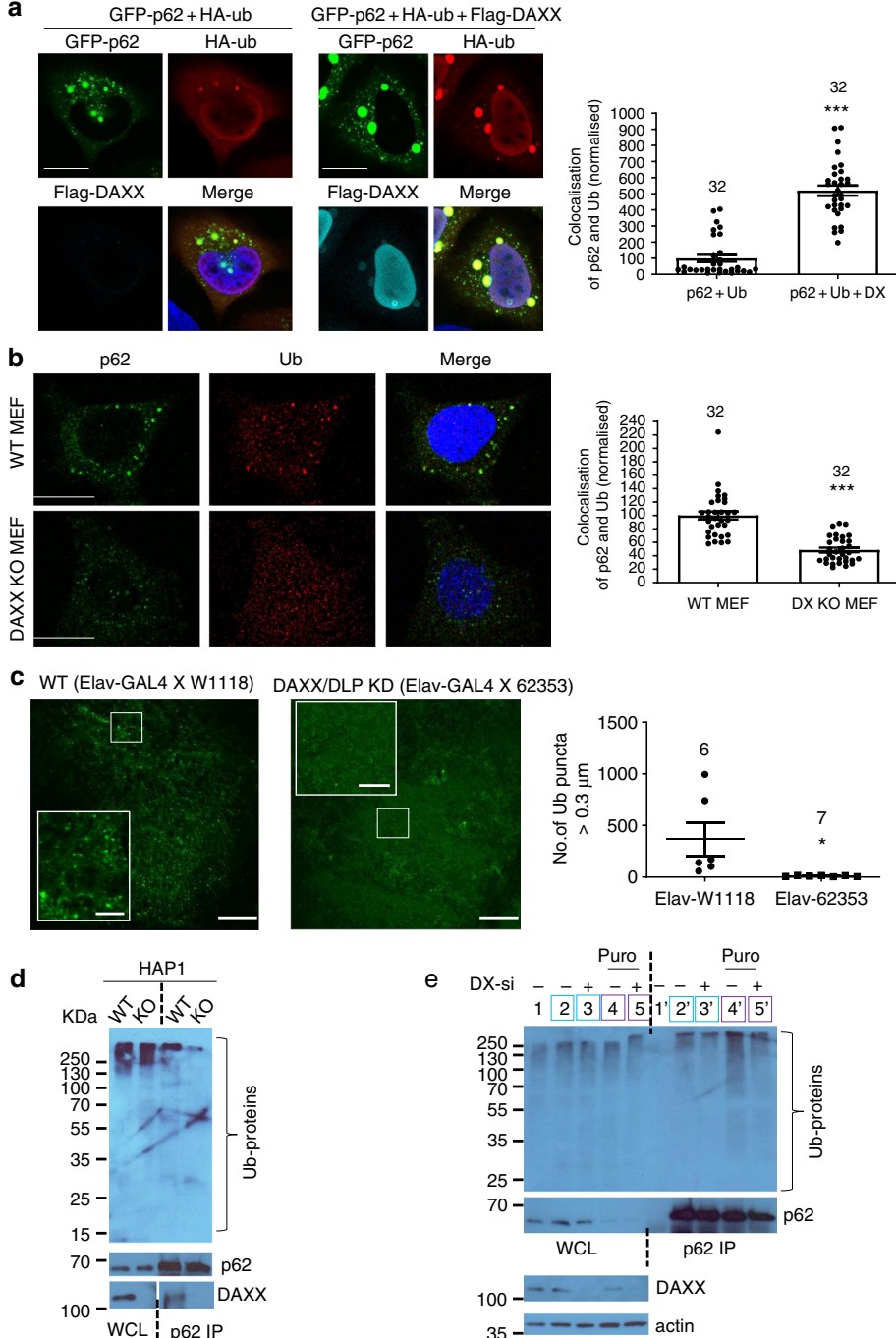

often exhibit multilayered structures with partially liquid- and solid-like characters, and in live cells, these condensates may be subject to various regulation. The regulation of protein oligomerisation would be one of the key factors for protein phase separation. Previous reports[29,41] as well as our current data suggest that p62 alone undergoes basal phase separation. We show here that DAXX regulates p62 oligomerisation and promotes p62 phase separation. Non-membrane compartments may be composed of hundreds of proteins, and very few proteins are predicted to promote liquid droplet formation[30,31]. In p62 puncta formation, it is yet unknown if these compartment protein components, in addition to DAXX and ubiquitinated autophagic substrates, can be a driving force for p62 phase separation. It would be interesting to experiment if other factors could also drive p62 phase separation.

DAXX as a histone H3.3 chaperone envelopes histone H3.3 and H4 for their chromatin loading[37]. DAXX is also known to interact with PML and co-localise with PML nuclear bodies[38]. It could generally promote liquid phase separation and condensation in these assemblies. DAXX is a major chaperone for the genomic deposit of the replication-independent histone H3, H3.3[67,68]. This would be important to maintain heterochromatin silencing and stability[67,69]. It would be interesting to investigate if the p62-DAXX interaction could affect the deposit of H3.3 and heterochromatin stability.

DAXX is critical to facilitate p62 in recruitment of ubiquitinated proteins and its downstream factors. p62 bodies recruit Keap1 to prevent Keap1-mediated Nrf2 degradation, allowing Nrf2 translocation into the nucleus to induce the expression of antioxidant enzymes[23,24]. Our study suggests that DAXX-induced

**Fig. 7** DAXX is required for p62 recruitment of polyubiquitinated cargos. **a** GFP-p62/HA-ubiquitin/vector or GFP-p62/HA-ubiquitin/Flag-DAXX were transfected into HeLa cells. After 20 h, the cells stained with anti-Flag and anti-HA antibodies. Confocal images were acquired. Bar: 10 μm. The co-localisation between GFP-p62 and HA-ubiquitin, in the absence or presence of Flag-DAXX, was quantified by the co-localisation area from LAS line profiling of the four largest co-localised puncta in each cell. Data were normalised with the mean of controls set as 100. $n = 32$ cells (each group) from three independently plated wells. Statistical analysis was performed with unpaired/two-tailed T-tests. Data are shown as mean ± sem. ***$P < 0.0001$. **b** WT and DAXX KO MEFs were stained with p62 and ubiquitin antibody. Confocal images were acquired. Bar: 10 μm. The p62/ubiquitin co-localisation was quantified by the co-localisation area from LAS line profiling of the largest co-localised puncta in each cell. $n = 32$ cells (each group) from three independently plated wells. Data were normalised as above. Statistical analysis was performed with unpaired/two-tailed T-tests. Data are shown as mean ± sem. ***$P < 0.0001$. **c** The *UAS*-control shRNA or *UAS*-DAXX shRNA *Drosophila* were crossed with *elav*-GAL4 flies to induce control or DAXX knockdown in the nervous system. The brains of the flies with DAXX-shRNA expression were dissected and stained with ubiquitin antibody. Bar: 20 μm; bar (insets): 5 μm. The sizes of ubiquitin puncta were quantified by ImageJ. Statistical analysis was performed with unpaired/two-tailed T-tests. $n = 6$ (WT), 7 (KD) biologically independent samples. Data are shown as mean ± sem. *$P = 0.0348$. **d** WT or DAXX KO HAP1 cells were lysed. The cell lysates were subjected to anti-p62 IP. The WCL and immunoprecipitates were probed with anti-ubiquitin, anti-p62, and anti-DAXX antibody. **e** HeLa cells were transfected with control siRNA or DAXX siRNA. HeLa cells with control siRNA or DAXX siRNA were non-treated (lane 2, 3/lane 2′, 3′); HeLa cells with control siRNA or DX siRNA were treated with puromycin for 5 h (lane 4, 5/lane 4′, 5′). The cell lysates were subjected to anti-p62 IP. The immunoprecipitates and WCL were subjected to SDS-PAGE, and immunoblot with anti-ubiquitin and anti-p62 antibody

p62 phase condensation is critical for Keap1 recruitment, promoting Nrf2-mediated stress response. DAXX has been found to be required for cell survival[34], but its prosurvival function is not well understood. In addition to a mechanism to p62 phase separation/condensation, this study offers mechanistic insight into the prosurvival function of DAXX. Mutations in both DAXX and Keap1-Nrf2 pathways are associated with cancers[70]. This study suggests a avenue to the association of DAXX mutations in tumorigenesis.

## Methods

**Antibodies and reagents**. The indicated antibody dilutions were used for western blot (otherwise indicated). Rabbit polyclonal antibodies: anti-p62 (1:3000) (MBL, PM045); anti-Flag (1:1000) (CST, #14793); anti-GFP (1:1000) (Abcam, ab6556); anti-actin (1:1000) (Sigma, A2066); anti-LC3 (1:3000) (CST, #12741); anti-ATG5 (1;1000) (Sigma, A0731); anti-DAXX (1:500) (Santa Cruz, sc-7152); anti-GST (1:10,000) (Sigma, G7781); anti-Myc (1:1000) (Sigma, C3956); anti-Nrf2 (1:1000) (MBL, PM069); anti-Lamin B1 (1:1000) (Abcam, ab16048). Mouse monoclonal antibodies: anti-GAPDH (1:5000) (Ambion, AM4300); anti-HA (1:1000) (Biolegend, 901501); anti-Flag (M2) (1:3000) (Sigma, F3165); anti-tubulin (1:5000) (Sigma, T5168), anti-Myc (9E10) (1:1000) (Sigma, M4439); anti-p62 (1:1000) (BD, #610833); anti-ubiquitin (1:500) (CST, #3936); anti-Keap1 (1:2000) (Origene, TA501989); anti-Nqo1 (1:1000) (CST, #3187); anti-GSTM1 (1:100) (DSHB, CPTC-GSTMu1-1); anti-Flag M2-agarose affinity gel (Sigma, A2220); Guinea pig polyclonal antibodies: anti-p62 (1:2000) (Progen, GP62-C). Dynabeads protein G (#10004D) for immunoprecipitation was purchased from Life Technologies. Doxycycline Hydrochloride (#10592-12-9) was from Fisher Bioreagents; puromycin (#53-79-2) was obtained from Santa Cruz. Cycloheximide, Chloroquine and Hygromycin B were from Sigma. G418 was a product of Thermo Fisher. The plasmids used for this study are listed in the Supplementary Table 1.

**DNA constructs**. Plasmids generated in this study are listed in Supplementary Table 1. Flag-DAXX (#27974), pCE-BiFC-VC155 (#22020), pCE-BiFC-VN173 (#22019) and LentiCRISPRv2 (#52961) were from Addgene. HA-DAXX was a kind gift from Dr Hsiu-Ming Shih. pMXs-puro-GFP-p62 K7A D69A was from Addgene (Plasmid #38281) gifted by N. Mizushima. p62 or DAXX point mutants were generated with the Quikchange Multi Site-directed Mutagenesis kit according to manufacturing instruction (Agilent Technologies, #200514).

**DNA and siRNA transfection**. HeLa cells were split 1 day prior to transfection to 50% confluence and left overnight in antibiotic-free DMEM containing 10% FBS. siRNAs were purchased from the suppliers as indicated. Human siRNA sequences: control siRNA-1 (Eurofins): 5′-CGUACGCGGAAUACUUCGA-3′; DAXX siRNA (Eurofins): 5′-GGAGUUGGAUCUCUCAGAA-3′; Atg5 siRNA (CST: #6345): 5′-GCCUGUAUGUACUGCUUUA-3′; Beclin 1 siRNA (CST: #6222): 5′-GGUCU AAGACGUCCAACAA-3′; Atg10 siRNA (Invitrogen): 5′-CCAUGGGACACU AUUACGC-3′; p62 siRNA (Eurofins): 5-GCATTGAAGTTGATATCGAT-3′. Smartpool Atg16L1 siRNAs (Dharmacon, L-021033-01): 5′-UGUGGAUGA UUAUCGAUUA-3 (Set 1)′; 5′-GGCACACACUCACGGGACA-3′ (Set 2); 5′-GCAUUGGAUUAACGGAAAC-3′ (Set 3); 5′-GUUAUUGAUCUCCGAACAA-3′ (Set 4). DNA constructs and siRNAs were transfected with Lipofectamine 2000 according to the manufacturer's instructions (unless otherwise stated). siRNAs were transfected at a final concentration of 50 nM. HeLa cells were maintained in 10% FBS DMEM containing no antibiotics for 48 h after transfection. TransIT-2020 (Mirus) was used for the transfection of MEFs and HAP1 cells.

**Cell culture**. The following cells were used for this study: Hela (ATCC, #CCL-2), HEK293T (ECACC, 12022001), Hela Tet-on p62-GFP (generated in this study), Atg7 KO and control MEFs (provided by M. Komatsu), Atg5 KO and control MEFs (provided by N. Mizushima), p62 KO and control MEFs (provided by M. Komatsu), DAXX KO and control MEFs (generated in this study), DAXX KO and control HAP1 (generated by Horizon Discovery) and Hela Tet-on DAXX (generated in this study). Authentication was confirmed by the suppliers, and by morphology check with light microscopy. All the cell lines used were negative for mycoplasma. HeLa cells and MEFs were cultured in Dulbecco's modified Eagle's medium (DMEM) supplemented with 10% FBS (Sigma). p62-GFP stably expressing Tet-on HeLa cells were cultured in DMEM supplemented with 10% FBS (Sigma) containing 100 μg/ml G148 and 50 μg/ml Hygromycin B. For expression of p62-GFP, the cells were induced with Doxycycline (250 ng/ml) for 20 h for p62-GFP expression. HAP1 cells were cultured in IMDM with 10% FBS. Where applicable, cells were treated with puromycin (10 μg/ml) or chloroquine (20 μM) for the indicated times.

**Generation of p62-GFP-inducible HeLa Tet-on cells**. pTRE-2Hyg-p62-GFP was transfected into HeLa Tet-on cells, and 300 μg/ml Hygromycin B was used to select cells resistant to the antibiotics. After 3 weeks, single clones were selected and 250 ng/ml doxycycline (Dox) was added to the cells to select GFP-positive clones. The positive clones were amplified for later experimental uses.

**Generation of CRISPR knockout plasmids**. For DAXX knockout MEFs, we generated a CRISPR/CAS9 editing plasmid by designing the following primers for the guide RNA pair. Forward: 5′-CACCGTGCTTAGTCCCACCCCGTCG-3′; reverse: 5′-AAACCGACGGGGTGGGACTAAGCAC-3′. To generate Lenti-CRISPRv2-gRNA, the oligos were subject to phosphorylation: 1 μl forward oligo (100 μM), 1 μl reverse oligo (100 μM) using T4 PNK in 10 μl reaction, and the oligos were annealed as follows: 37 °C for 30 min, 95 °C for 5 min and ramp down to 25 °C at 5 °C per minute. Bsmb1/Esp31-digested pLentiCRISPRv2 was ligated with annealed oligos (1:200, 1 μl). Positive plasmids were confirmed by DNA sequencing.

**Lentivirus production and generation of DAXX knockout MEFs**. For Lentivirus production, HEK293T cells were plated in a 6-well plate for 80% confluence on the next-day transfection. HEK293T cells were transfected with 5 μg of DNA per well in a ratio of 1:3:4 (pCMV-VSVg:psPAX2:pLentiCRISPRv2-DAXXgRNA) with 10 μl Mirus 2020. Viruses were harvested by medium collection after 24 h, with two repeats of 24-h interval.

MEFs were infected with the CRISPR/CAS9 system-derived viruses containing DAXX knockout or control element. Transfected cells were selected with puromycin (5 μg/ml). After 4 weeks, single clones were selected. After amplification, western blot was used to verify the effectiveness of DAXX knockout.

**Generation of DAXX knockout haploid HAP1 cells**. HAP1 DAXX KO or control cells were generated by Horizon Discovery (Cambridge, UK) with the CRISPR/CAS9 gene editing approach. The guide RNA sequence (5′-CTAGGGTCCTGTCT CGGGCC-3′) was used to target human DAXX exon 3. DAXX knockout with early termination of protein translation was confirmed by DNA sequencing and western blot.

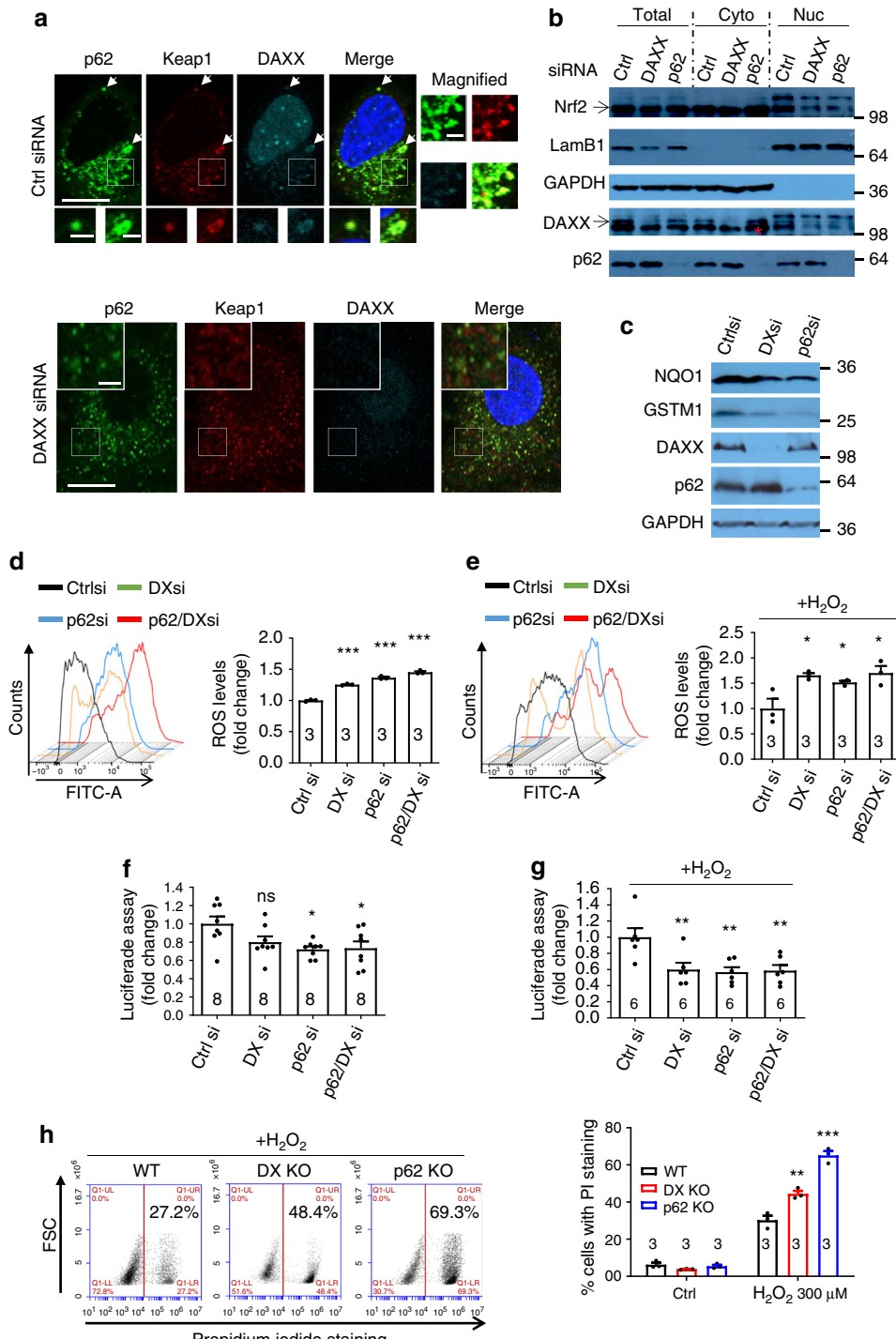

**Primer sequences for mutagenesis.** *DAXX mutants*

DAXX-R306Q: 5′-AGC CTT GGC CTC CCC CAA CAG CAG CTC CAG CTC-3′

DAXX-F317A: 5′-CAG GAT GCC GCC CGA GAT GTG-3′

DAXX-D349A: 5′-CCA GGC GTT GCC CCT GCA CTA-3′

*p62 mutants*

p62-P228L: 5′-GAATCAGCTTCTGGTCTATCGGAGGATCCGAGT-3′

p62-R321C: 5′-GAGTCCGAGGGGTGCCCTGAGGAAC-3′

p62-S370P: 5′-TCTCTGGACCCCCCCCAGGAGGGAC-3′

p62-A381V: 5′-GGGGCTGAAGGAAGCTGTCTTGTACCCACATCTC-3′

p62-P392L: 5′-CCAGAGGCTGACCTGCGGCTGATTGAG-3′

p62-M404V: 5′-TCCCAGATGCTGTCCGTGGGCTTCTCTGATG-3′

**Yeast two-hybrid screening.** Clontech Matchmaker Gold Yeast Two-Hybrid System was used for our yeast two-hybrid (Y2H) screening. Briefly, p62 1–300aa as a bait was cloned into pGBKT7. The bait cDNA was transformed into Y2HGold (the yeast strain). cDNA expression was confirmed by in vitro translation. The yeast Y187 harbouring human cDNA library (Mate & Plate Human Heart Library, #630417) was mated with Y2HGold with p62 1–300aa. Following the mating, four reporters detect protein–protein interactions, namely Aureobasidin A (Aba) resistance, histidine (His) minus resistance, adenine (Ade) minus resistance and blue colour with X-alpha-Gal. In addition, pGBKT7 plasmid confers leucine (Leu) minus resistance and pGADT7 confers tryptone (Trp) minus resistance. Therefore, media with Leu-, Trp- and/or His-, Ade- along with Aba resistance and blue colour were used to select positive yeast clones. Plasmids were isolated from the positive

**Fig. 8** DAXX activates Nrf2-mediated stress response. **a** HeLa cells were treated with control or DAXX siRNA for 48 h. Cells were fixed and stained with anti-p62, anti-Keap1 and anti-DAXX. White arrowheads mark condensed signals, which are magnified at lower panels; boxed areas are magnified at right panels. Bar: 10 μm; bar (inset/magnified): 2 μm. **b** HeLa cells were knocked down with control, DAXX or p62 siRNA for 48 h. The total lysates, cytoplasmic (Cyto) or nuclear (Nuc) fractions were used for immunoblot with anti-Nrf2, Lamin B1 (LamB1), DAXX, p62 or GAPDH antibody. The red star denotes the remaining Nrf2 signals from the previous blot. Lamin B1: nuclear marker. **c** HeLa cells were knocked down with control, DAXX or p62 siRNA, as indicated, for 48 h. Cells were lysed and subjected to immunoblot with indicated antibodies. **d, e** HeLa cells were knocked down with siRNAs as indicated for 48 h, and subsequently treated with control (**d**) or $H_2O_2$ (300 μM) (**e**) for 16 h. Cells were subjected to $H_2DCFDA$ (DCF) staining, and flow cytometry analysis. Statistical analysis was performed by one-way ANOVA. Tukey's test was used for the comparison. $n = 3$ independently plated/treated samples for each group. Data are shown as mean ± sem. ***$P < 0.0001$; *$P < 0.05$. **f, g** HeLa cells were knocked down with siRNAs as indicated, and subsequently transfected with luciferase reporter components. Cells were treated without (**f**) or with $H_2O_2$ (200 μM, 6 h) (**g**). Cell lysates were subjected to dual luciferase assays. $n = 8$ independently plated wells for each group (**f**); $n = 6$ independently plated wells for each group (**g**). Statistical analysis was performed by one-way ANOVA. Tukey's test was used for the comparison. Data are shown as mean ± sem. *$P < 0.05$; **$P < 0.01$. **h** Wild-type, DAXX knockout, or p62 knockout HAP1 cells were treated with $H_2O_2$ as indicated. After 20 h, the cells were subjected to propidium iodide staining, and flow cytometry for cell death analysis. $n = 3$ independently plated/treated samples for each group. Statistical analysis was performed by one-way ANOVA. Tukey's test was used for the comparison. Data are shown as mean ± sem. **$P < 0.01$; ***$P < 0.001$

yeast clones, and pGADT7 cDNAs were recovered by transforming them into *E. coli* with LB media containing kanamycin (50 μg/ml). DNA sequencing and gene BLAST were performed successively. pGBKT7-p62 1-300aa-Y2HGold and pGADT7 1-370aa-Y187 were mated and the yeasts were subsequently cultured in media with Aba, X-alpha-Gal, His minus and Ade minus, to confirm the direct interaction between p62 1-300aa and DAXX 1-370aa in yeast.

**Cell fraction of NP-40 and urea**. $10^6$ x transfected HeLa cells were resuspended in 100 μl Buffer A (20 mM Tris-HCl, pH 7.4, 2 mM $MgCl_2$, 0.5% NP-40) with protease inhibitor cocktail (Roche) on ice for 15 min. Thirty microlitres of cell lysate was taken as a total cell lysate. The rest of the total cell lysate was subjected to 17,000 × *g* centrifuge for 10 min. The supernatant was kept as NP-40 fraction. The pellet was dissolved in 60 μl 8 M urea-containing Buffer A on ice for 20 min. The total cell lysate, NP40 fraction and urea fraction were mixed with equal volumes of 2× Laemmli buffer. Fifteen microlitres of samples were used for SDS-PAGE and subsequently immunoblot.

**In vitro translation**. In vitro translation was performed in TNT-coupled reticulocyte lysate systems (L4610, Promega) following Promega instruction. Briefly, 1 μg of pcDNA3-DAXX or pcDNA3-p62 was combined with the components as required: TNT reticulocyte lysates, reaction buffer, RNA polymerase, amino acid mixture and RNasin ribonuclease inhibitor. Fifty microlitres of the mixture was incubated at 30 °C for 90 min. In total, 0.5 μl was taken for western blot analysis.

**In vitro binding assays**. We performed in vitro binding assays to test a direct interaction between p62 and DAXX. Two-way pull-down experiments were tested: (1) 2 μg glutathione beads-bound GST or 2 μg glutathione beads-bound GST-p62ΔPB1 was incubated with 5 μl of in vitro-translated DAXX for 3 h at 4 °C. The pull-down products were subjected to SDS-PAGE and immunoblot with DAXX antibody; (2) 2 μg glutathione beads-bound GST, or 2 μg glutathione beads-bound GST-DAXX 1-250aa was incubated with 5 μl of in vitro-translated p62 for 3 h at 4 °C. The pull-down products were subjected to SDS-PAGE and immunoblot with p62 antibody.

**Immunoprecipitation**. Immunoprecipitation (IP) was performed using Buffer A (20 mM Tris-HCl, pH 7.4, 2 mM $MgCl_2$, 150 mM NaCl, 5 mM NaF, 1 mM $Na_3VO_4$, 0.5% NP-40, protease inhibitor cocktail (Roche)). Cells were lysed in Buffer A for 20 min on ice, followed by centrifugation at 13,000 × *g* for 15 min. In total, 500 μg–1 mg total protein were used as the starting material for IPs. An antibody (or anti-Flag M2-agarose affinity gel) was added to a final concentration of 5 μg/ml and incubated for 2 h to overnight at 4 °C. IP products were directly boiled in Laemmli buffer and subjected to PVDF membrane transfer and western blot.

**Immunocytochemistry**. Immunocytochemistry was performed as previously described. Cells were fixed with 4% paraformaldehyde for 10 min. The fixed cells were washed three times in PBS, then permeablised with 0.5% Triton in PBS for 10 min. Cells were blocked in blocking buffer (1% BSA, 1% heat inactivated goat serum in PBS) for 30 min at room temperature. Primary antibodies were incubated with cells overnight at 4 °C. The secondary antibody was incubated for 30 min after washing three times (10 min, each). Cells were washed three times (10 min, each) after incubation with secondary antibodies, then mounted with DAPI (1 μg/ml). Images were acquired on a Leica confocal microscope.

**Production of GST-p62ΔPB1 and GST-DAXX 1-250 in *E. coli***. p62ΔPB1 (Δ1-120aa) or DAXX 1-250aa was subcloned into pGEX-6P-1 (Amersham) for

recombinant GST-p62ΔPB1 or GST-DAXX 1-250aa. The plasmids as well as the empty plasmid pGEX-6P-1 were then transformed into BL21 (DE3). A final concentration of 0.2 mM IPTG was added to culture media (LB) to induce recombinant protein expression.

GST-tagged proteins (or GST only) were purified with glutathione-sepharose (GE Healthcare). Briefly, pellets were resuspended in PBS containing 1% Triton X-100 and protease inhibitor cocktail (Roche) and 0.1 mg/ml PMSF, then lysed by sonication. The lysates were subjected to 20,000 × *g* centrifugation for 30 min. The supernatants were incubated with glutathione-sepharose beads for 2 h. The glutathione beads were washed with PBS and eluted with 10 mM of glutathione in 50 mM Tris (pH 8.0). Eluates were dialysed against binding buffer A (150 mM NaCl, 20 mM Tris, 2 mM $MgCl_2$, pH 7.4).

***Drosophila* models**. The *Drosophila* lines were obtained from the Bloomington *Drosophila* Stock Center at University of Indiana (http://flystocks.bio.indiana.edu/), and maintained in a 25 °C incubator. The male flies with WT background (W1118) or *UAS*-DAXX shRNA (62353) were crossed with virgin female flies of the nervous system driver line *elav*-GAL4 (*c155*) to generate the desired genotypes for DAXX knockdown or control in nervous system.

**qRT-PCR analysis**. The quality and quantity of RNA were measured with a Nanodrop. Total RNAs with 260/280 ratio >1.8 and a 260/230 ratio >2.0 were used for subsequent qPCR analysis. RNA was reverse transcribed using the High-Capacity cDNA Reverse Transcription Mix (Applied Biosystems) using cycles of 25 °C (10 min), 37 °C (120 min) and 85 °C (5 min). cDNA templates were then used for qPCR using a LightCycler 480 DNA SYBR Green I Master kit (Roche) in the LightCycler 480 II system (Roche). The reaction conditions were as follows: Denaturation (95 °C for 5 min), amplification was performed for 45 cycles (95 °C for 10 s, 60 °C for 20 s, 72 °C for 10 s) and a melt curve programme (65–95 °C with a heating rate of 0.1 °C/s). qPCR was performed in triplicate for each sample and a relative quantification approach was performed using the $2^{-\Delta\Delta Ct}$ method.

*Mammalian cells*. RNA was isolated with RNAeasy kit as instructed (Qiagen). Homogenisation of the tissue was undertaken using a syringe and 0.8 mm gauge needle (BD Plastipak). One microgram of total RNA was used for reverse transcription. Three samples per condition were used for qPCR analysis. All primers were from Sigma and used at 0.5 μM. Actin was used as a control to normalize the data. Human actin primers: 5′-ACTGGCATCGTGATGGACTC-3′ (forward) and 5′-TCAGGCAGCTCGTAGCTCTT-3′ (reverse); human NQO1 primers: 5′-AAAGGACCCTTCCGGAGTAA-3′ (forward) and 5′-CTTGGAA GCCACAGAAATGC-3′ (reverse); human GSTM1 primers: 5′-TGTCCTT GACCTCCACCGTA-3′ (forward) and 5′-GAACACAGGTCTTGGGAGGA-3′ (reverse).

*Drosophila*. Fly brains were dissected. The brains from eight flies per genotypes (at the age of 7 days) were pooled for each RNA sample. Three total RNA samples from flies per genotype were used for qPCR analysis. Homogenisation of the tissue was undertaken using a syringe and 0.8 mm gauge needle (BD Plastipak). Total RNA was extracted using RNeasy Plus Mini kit (Qiagen) according to manufacturer's instruction. In all, 0.5 μg total RNA was used for reverse transcription. Primers were from Sigma and used at 0.5 μM, and alpha-actin was used as an internal control. Fly DLP (DAXX) primers: 5′-CCCAATACTGC AGTGGCTCA-3′ (forward) and 5′-CACGGGTCTAAGTGGCTGTT-3′ (reverse). Actin primers: 5′-GCGTCGGTCAATTCAATCTT-3′ (forward), 5′-AAGCTGCAA CCTCTTCGTCA-3′ (reverse).

**Immunofluorescence (tissues)**. Briefly, tissues were blocked with 10% goat serum in TBS with 0.1% Triton X-100 (TBS-T) for 1 h at RT. Primary antibodies were diluted in TBS-T and incubated with tissue sections overnight at 4 °C. Following 3-time (10 min each time) washing in TBS, secondary antibodies were applied to the

tissues for 30 min. After 3-time (10 min each time) washing, tissues were mounted with DAPI (1 μg/ml). Images were acquired with a confocal microscope.

**Proteasome activity assays**. Proteasome activity assays were carried out according to the manufacturer instruction (Abcam). Briefly, cells were harvested, and washed with cold PBS. The cells were resuspended in 0.5% NP-40 (~4 volumes), and lysed by pipetting up and down a few times. The cell lysates were subjected to centrifugation (10 min) at 4 °C at 13,000 rpm. The supernatants were collected. One microlitre of proteasome substrate was added to 50 μl cell lysate. The mixtures were incubated at 37 °C for 30 min, protected from light. The samples were measured with a fluorometric microplate reader at 350/440 nm.

**Nuclear fractionation**. The cells from a 35-mm dish were suspended and lysed in 300 μl Buffer A (20 mM Tris-HCl, pH 7.4, 2 mM $MgCl_2$, 150 mM NaCl, 5 mM NaF, 1 mM $Na_3VO_4$, 0.5% NP-40, protease inhibitor cocktail (Roche)) by pipetting, and incubated on ice for 15 min for lysis. The cell lysates were centrifuged at 3000 rpm ($720 \times g$) for 5 min. The pellets contained nuclei and the supernatants were cytoplasmic fraction. The nuclear fraction was washed with 300 μl Buffer A once and subjected to centrifugation at 3000 rpm for 5 min to collect the pellets.

**Mice**. Animal work performed in UK complied with UK Home Office legislation: the UK Animals (Scientific Procedures) Act with appropriate Home Office Project and Personal animal licences and with local Ethics Committee approval. The procedures using mice in Thomas Jefferson University were approved by the institutional animal care and use committee (IACUC) at Thomas Jefferson University. DAXX conditional knockout and control mice were produced and maintained at Thomas Jefferson University, as described previously[47]. DAXX f/f mice were crossed with *LckCre* transgeneic mice. The resulting heterozygous DAXX+/f *LckCre* mice were intercrossed to obtain DAXX f/f *LckCre* mice (T cell lineage DAXX deficient mice). DAXX f/f *LckCre* mice were subsequently crossed with DAXX f/f mice to produce DAXX f/f *Lck* mice and DAXX f/f control mice. Genomic DNA was extracted from the mouse tail. Wild-type and conditional DAXX knockout alleles were genotyped by PCR using primers 5′-AGCAGTAACTCCGGTAGTAGGAAG-3′ and 5′-AGGAACGGAACCACCTC AG-3′. The *LckCre* transgene was genotyped by PCR using primers 5′-CC GAAATTGCCAGGATCAGG-3′ and 5′-CTTACCTGTAGCCATTGCAGC TAG-3′.

**6xHis-tagged protein expression and purification**. p62 or DAXX was cloned into pET-28a (Novagen) for His-tagged protein expression. The plasmids were transformed into BL21 (DE3). A final concentration of 0.2 mM IPTG was added to culture media (LB) to induce recombinant protein expression.

6xHis-tagged proteins were purified with $Ni^{2+}$-charged 6xHis-tag affinity resins (Novagen) according to manufacture instruction. The pellets were resuspended in binding buffer (0.5 M NaCl, 20 mM Tris, 5 mM imidazole, pH 7.9) containing protease inhibitor cocktail and 0.1 mg/ml PMSF. The cells were lysed by adding 1/10 volume of Bugbuster 10× protein extraction reagent (Millipore) and 50 U/ml benzonase (Thermo). After centrifugation at $16,000 \times g$, 20 min, the supernatants were applied to 6xHis-affinity beads. The beads were washed with binding buffer and wash buffer (0.5 M NaCl, 20 mM Tris, 60 mM imidazole, pH 7.9) and eluted with elute buffer (0.5 M NaCl, 20 mM Tris, 1 M imidazole, pH 7.9). The proteins were subjected to size exclusive chromatography with Superose 6 Increase 10/300 GL (GE Healthcare). The purified proteins were concentrated to at least 10 μM using Amicon Ultra-0.5 mL Centrifugal Filters in the desired stock buffer as described below.

**In vitro phase separation assays**. For in vitro phase separation, purified recombinant p62 or DAXX was centrifuged at $16,000 \times g$ for 5 min to remove any potential protein aggregates. p62 droplet phase separation was carried out in a glass-bottomed well in a 384-well plate or a microcentrifuge tube, in the buffer containing 40 mM Tris-HCl pH 7.4, 150 mM NaCl, 1 mM DTT, at room temperature. Imaging was performed on a 384-well plate or a glass slide. p62 was stocked in the buffer containing 40 mM Tris-HCl pH 7.4, 300 mM NaCl, 1 mM DTT, 10% glycerol or the buffer containing 40 mM Tris-HCl pH 7.4, 1 mM DTT, 10% glycerol. Purified recombinant DAXX was stocked in the buffer containing 40 mM Tris-HCl pH 7.4, 150 mM NaCl, 1 mM DTT. For in vitro phase separation, the stock of p62 protein was adjusted to the phase separation buffer conditions. Images were acquired with Leica DMi8 microscopy or confocal microscopy SP8.

**In vitro sedimentation assays**. Purified recombinant p62 or DAXX was centrifuged at $16,000 \times g$ for 5 min to remove any potential protein aggregates, immediately prior to in vitro sedimentation experiments. In vitro p62 droplets were spun down at $12,000 \times g$ for 5 min, upon phase separation reaction. The pellet was washed with 500 μl phase separation buffer once, and subjected to immunoblot.

**Native PAGE for in vitro p62 oligomerisation**. Two microlitres of in vitro-translated p62 was incubated with purified 0.3 μg GST or GST-DAXX 1-250 in 1× PBS (20 μl) at room temperature for 48 h. The incubated mixtures were subjected to native PAGE and probed with p62 antibody.

**Simple Wes size assays (capillary-based electrophoresis)**. The methods were adopted according to the protocol provided by the manufacturer (Protein Simple/Biotechne, cat No: SM-W004). Briefly, cells were lysed in Buffer A (see the recipe above in IP method in supplementary information) with complete protease inhibitors (Thermo Scientific). The cell lysates were cleared by $13,000 \times g$ centrifugation. The lysates were diluted with an equal volume of distilled water. Protein concentrations were measured with a Nanodrop spectrophotometer (Thermo Scientific). The cell lysates were further diluted with 0.1× Sample buffer to achieve a protein concentration of 0.4 μg/μl. In total, 1.4 μl of 5× Master mix was added to 5.6 μl of diluted cell lysates. Five microlitres of the mixtures were loaded to the WES setup (Protein Simple/Biotechne, CA) for the Simple Wes assays. In total, 12–230 kDa separation capillaries were used in this work. Of note, all buffers were DTT-free, as the samples were run in native conditions. The antibody dilutions were applied, as indicated in each figure legend.

**Native PAGE**. One 35-mm-well cells were lysed in 100 μl Buffer A containing 0.5% NP-40 and protease inhibitor cocktail for 15 min. Twenty-five micrograms of each sample was used for 10% or 12% native SDS-PAGE (Bio-Rad) and immunoblot with the antibodies indicated.

**Measurement of the sizes of puncta and co-localisation**. The sizes of p62 (or ubiquitin/LC3) puncta were measured with the two software, Leica LAS AF lite or ImageJ (particle analysis), as indicated. The co-localisation was determined by the co-localisation area (length × amplitude) from LAS AF Lite line profiling of the four largest co-localised puncta in each cell.

**Nrf2 luciferase reporter assays**. Luciferase assays were carried out with transfections of ARE firefly luciferase reporter vector + constitutively expressing Renilla luciferase vector (BPS Bioscience) into HeLa cells knocked down with control siRNA, DAXX siRNA, p62 siRNA or DAXX + p62 siRNA. Briefly, HeLa cells were transfected with control siRNA, DAXX siRNA, p62 siRNA or DAXX + p62 siRNA. After 20 h, the cells were split into a 96-well plate. The reporter components (ARE luciferase reporter vector + constitutively expressing *Renilla* luciferase vector (internal reporter)) were transfected 20 h post the splitting. Dual luciferase assays were performed with Promega DLR kit according to the instruction.

**Cell death measurement**. Trypsinised cells were washed twice with cold PBS and resuspended in 1× binding buffer (10 mM HEPES, pH 7.4; 140 mM NaCl; 2.5 mM $CaCl_2$) at $1 \times 10^6$ cells/ml. Hundred microlitres of these cells were transferred to a FACS tube, treated with 5 μl propidium iodide (30 μg/ml) (Sigma) and incubated for 15 min at room temperature. Each tube had a further 200 μl of 1× binding buffer added and then was analysed with the Becton-Dickinson (BD) Accuri C6 flow cytometer.

**ROS measurement**. Cells were incubated with media containing 5 μM H2DCFDA (Thermo) for 20 min at 37 °C, then washed twice with PBS. Next cells were trypsinised, washed with flow fluid (PBS with 2% FBS), then resuspended in 500 μl of flow fluid. Flow cytometry was performed using BD FACSAria II cytometer. Data were analysed with FlowJo software.

**Fluorescence recovery after photobleaching**. HeLa cells or MEFs were transfected with GFP-p62 or GFP-p62/mCherry-DAXX. After the times as indicated, the live cells were subjected to FRAP analysis using an oil immersion objective (×40). GFP-p62 puncta were bleached for 11.8 s (8 × 1.477 s) with a 100% laser intensity of 488 nm. Recovery was recorded by fluorescence intensity for the indicated times.

**Quantification of autoradiographs**. To quantify protein band density, the relevant specified bands were analysed using ImageJ software. The relative value was computed.

**Statistical analysis**. Statistical analysis was performed primarily with Graphpad Prism. The unpaired two-tailed *t*-tests were conducted for the comparison between two groups; one-way or two-way ANOVA tests were used for the comparison among multiple groups: one-way ANOVA for variables influenced by a single factor; two-way ANOVA for variables influence by two or more factors (***$P < 0.001$, **$P < 0.01$, *$P < 0.05$, ns, not significant). In the cases where data were processed with the Excel, *t*-tests were used and *P*-values were determined by unconditional logistical regression analysis by using the general loglinear option of SPSS 9.1 (***$P < 0.001$, **$P < 0.01$, *$P < 0.05$, NS, not significant). Data from at least three independent experiments were analysed.

**Full blot images**. Full blot images are shown for important blots in Supplementary Fig. 13.

## Data availability
The data that support the findings of this study are available from the corresponding author upon reasonable request.

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

## Acknowledgements

We are grateful to the Medical Research Council UK (MR/M023605/1) (S.L.), the National Natural Science Foundation of China (31428014) (S.L.), BRACE Charity (S.L.), The UK Academy of Medical Sciences and Newton Fund for Newton Advanced Fellowship (NAF/R1/191045) (S.L. and B.L.), the National Natural Science Foundation of China (91649105) (B.L.), the National Key Research and Development Program of China (2016YFC0905100) (B.L.), and Biotechnology and Biological Sciences Research Council UK (BB/L02392X/1) (B.H.) for funding. We are grateful to Dr Masaaki Komatsu (Niigata University) for kindly providing us with ATG7 WT/KO MEFs and p62 WT/KO MEFs. We are thankful to Dr Torsten Bossing for excellent support in imaging.

## Author contributions

S.L. conceived, initiated the project and designed the experiments. Y.Y., T.W. and S.L. performed the experiments including Y2H, immunoprecipitation, immunocytochemistry, immunofluorescence, immunoblot, p62 puncta/oligomerisation assays, qPCR and FRAP; Y.Y. and S.L. performed recombinant protein purification and in vitro phase separation. R.B. contributed to FRAP and cell death analysis; C.S. performed BiFC experiments, and contributed to immunocytochemistry and immunoblot; Y.F. contributed to immunostaining; X.W. contributed to fly tissue assays; P.G. and B.H. contributed to Wes assays; T.E. contributed to tissue sectioning; Y.F. helped with immunoblot; S.R. helped with immunostaining; R.S. helped with cell culture; J.Z. provided the thymus of control and DAXX cKO mice. S.L. and B.L. analysed the data. S.L. wrote the paper.

## Additional information

**Competing interests:** The authors declare no competing interests.

