## [Peer Review File · Nature Communications]

Reviewers' comments:

Reviewer #1 (Remarks to the Author):

In this report, Yang et al. identify DAXX as a novel p62-interacting protein. DAXX mediates oligomerization of p62 in a PB1-independent manner, which drives p62 phase separation. DAXX-p62 condensates further recruit Keap1, leading to dysregulation of Nrf2 responses.

These results are novel and timely. However, as described in detail below, this study relies heavily on over-expression experiments and the role of endogenous DAXX is determined less robustly. Several different DAXX cell lines and RNAi-mediated knockdown were used, which makes this study rather complicated. Also, some in vitro studies would be required to conclude that DAXX indeed promotes phase separation.

Major comments:

1. It could be true that DAXX interacts with p62 and promotes phase separation in vivo. However, these data do not necessarily indicate that DAXX itself promotes phase separation of p62. For example, ubiquitin or other proteins may mediate this process. To prove the authors' hypothesis, some in vitro studies are required. Does purified DAXX actually promote p62 droplet formation in vitro? Without such data, the conclusion must be amended.
2. The role of endogenous DAXX in p62 droplet formation should be investigated more vigorously and consistently using knockout cells. The authors use HAP1 DAXX KO cells as a negative control in Figures 2 and 4 and DAXX KO MEFs in Figure 6. The results seem to be much clearer in KO MEFs than in KO HAP1 cells as there is only a small difference between WT and HAP1 DAXX KO cells in Figure 2c. The authors also use knockdown cells in some other experiments. It is highly recommended that the authors consistently use the same cell line (ideally KO MEFs) as a negative control throughout the study so that the experiments in different figures can be directly compared.
3. The physiological importance of DAXX to formation of p62 bodies is not very clear. Do DAXX KO MEFs demonstrate smaller p62 droplets after treatment with puromycin (as in Figure 5), proteasome inhibitors, autophagy inhibitors (such as wortmannin), or arsenate?

4. Figure 5a: The authors conclude that the PB1 domain is not required for oligomer formation of p62 by DAXX. However, the formation of condensates (Fig. 5a) and oligomers (Fig. 5b) of p62dPB1 induced by DAXX appears to be much milder than those of WT p62 (Figs. 1, 2, and 4). Rather, these results suggest that the PB1 domain is still important for DAXX-induced oligomer formation. It is also possible that the effect of PB1 deletion might have been overlooked missed because the PB1 deletion mutant possibly interacts with endogenous p62 through the UBA domain (PMID: 21715324). Thus, the p62dPB1 mutant should be introduced into p62 KO cells. Also, to test the role of PB1 domain, the K7R D69A mutant should be included.

5. To determine the significance of p62-DAXX interaction, the authors use only deletion mutants in vivo. However, as the deleted regions are too large, it may affect intrinsic functions other than binding with p62. The authors should narrow down the region required for the binding and use point mutants in the phase separation and Nrf2 activity experiments.

6. In Figure S3, why does the size of p62 puncta not increase in various Atg knockdown cells? This is contradictory to previously published results in many Atg knockout cells. The protein level of p62 should be checked in these knockdown cells to confirm that these Atg proteins are indeed depleted.

7. Figure 3: The data suggest that DAXX induces more gel-like structures rather than liquid droplets. This should be mentioned.

8. The mRNA and protein levels of p62 should be determined in DAXX KO and DAXX overexpressed cells to confirm that the effect does not result from transcriptional alteration.

9. Fig. 1d needs a negative control.

Minor comments:

1. In Introduction or the first paragraph of Result, it should be mentioned that previous studies suggest that ubiquitin can promote phase separation or droplet formation of p62 (Sun et al. Cell Res 2018 (ref #31), Zaffagnini et al. EMBO J. 2018).

2. Figure 2: The expression level of p62 and DAXX should also be determined by immunoblotting. Ideally, the Tet system should be used to express DAXX instead of p62 if the authors want to see the effect of DAXX on p62 droplet formation.

3. Figure 4b: To prove that the GFP fragment is a result of autophagic degradation, ATG knockout or knockdown cells should be used.

Reviewer #2 (Remarks to the Author):

This manuscript builds on the previous extensive studies of the authors on the role and regulation of autophagy. In this study, however, they have identified the role of cytoplasmic DAXX in driving p62 liquid phase condensation by inducing its oligomerisation. Furthermore, they show the role of DAXX mediated p62 phase condensation as a mechanism of stress adaptation acting via the Nrf2 pathway.

Recent studies emphasized the concept that stress adaptation through protein phase separation or open macromolecular assemblies is a widespread phenomenon in nature. Phase separating proteins including p62 are associated with age related diseases and phase transition is proposed to drive progression of such diseases. In this context the current manuscript provides interesting insights into this biological phenomena.

However the authors should discuss and address some critical concerns:

1. The study is based mostly on two cell lines. Considering the conflicting literature on Daxx localization (nuclear vs cytoplasmic), it is critical to assess the proposed interactions between Daxx and p62 in more physiological systems and ideally also in tissues (at least by immunofluorescence). In this respect, access to tumor tissues displaying loss of Daxx expression (LoF mutations are found in a number of human neoplasms) would allow the authors to correlate loss of Daxx to alterations of p62 distribution in disease-relevant settings. Finally, it is surprising that the authors have not discussed how interaction between Daxx and a p62/Nrf2 pathway might affect Daxx H3.3 chaperone activity.

2. Foremost, the p62 deficiency has been proposed to enhance proteasome activity, which degrades polyubiquitinated proteins (Korolchuk et al., 2009; Moscat and Diaz-Meco, 2009), hence it critical to evaluate the effect of the DAXX mediated p62 liquid phase condensation on the half life p62 itself and on the proteosomal activity and polyubiquitinated proteins levels over time in these cells.

3. To further increase the impact of this study, authors should look into the effect of P62 mutations has been identified in Paget's disease of the Bone and ALS disease on Daxx interaction and its ability to affect p62 homeostasis. While most such mutations lie in the polyubiquitin chain binding UBA domain and not in the PEST domain identified by the authors as the P62 domain interacting with DAXX, its is pertinent to evaluate if DAXX mediated p62 body formation is altered by such mutations and this could add important details to the as of yet unclear mechanism behind these mutation related pathologies.

4. While all p62 puncta colocalizes with DAXX puncta, not all DAXX puncta are positive for p62, the authors to discuss this interesting disparity.

Minor comments:

1. The haploid HAP1 cells has been shown to have elevated levels of p53, a known DAXX interacting protein and might influence the assays with DAXX. Hence it would be important to perform the KO assays in diploid cells.

2. It is difficult to appreciate the Fig.2 c claiming that "The formation of p62 puncta is weakened in DAXX knockout HAP1 cells", as the puncta like structures are present in the DAXX KO HAP1 cells as well.

3. The quality of the gel image makes it impossible to appreciate the Fig. 6, d, claiming "p62-polyubiquitinated proteins interaction is weakened in DAXX-KO HAP1 cells" and the image quality is as such below the Journal publication standards.

4. Quality of WB data in Figure 3e, 4b and 5c is below standard and makes conclusions related to these data not very convincing.

5. It would be important to provide details on the efficiency and level of DAXX KD in the UAS-DAXX shRNA drosophila experiment to evaluate the correlation with decrease in ubiquitin puncta as proposed in the Fig 6, c.

6. The Thymus has been misspelled twice in the Author Contributions (page 30) and Fig 2, d. legend.

Reviewer #3 (Remarks to the Author):

The manuscript by Yang et al identifies the protein DAXX as a binding partner of p62. The authors aim to test the effect of DAXX on p62 phase separation and function. While they generate interesting data, their interpretation that p62 undergoes phase separation and DAXX enhances it, is not justified. While the insight that DAXX is a player in the p62/Keap/Nrf2 signaling axis is interesting and important, the work is overall too preliminary for publication in this reviewer's opinion.

My major comments are as follows:

1. The authors do not demonstrate that the system they study actually undergoes liquid-liquid phase separation. They show punctate structures in the cytoplasm of cells. If we consider that p62 phase separation has recently been demonstrated by Su et al. (ref 31 in this manuscript), we would still need to see evidence that DAXX enhances phase separation of p62. We only see more/larger punctate structures in the presence of DAXX.
2. To compare different conditions, the authors measure the size of the largest punctum in cells and the percentage of cells with particularly large puncta (> 1 μm) as a surrogate for the volume fraction of the "dense phase" (i.e. the p6 bodies). It would be far superior to measure the actual volume fraction (i.e. the total volume of bodies compared to the total volume of cells). If the authors do not do this, they should at least discuss that their measure is a suboptimal surrogate of what they should really be measuring.
3. Possible explanations for observing a larger volume fraction of p62 bodies are the following: (a) DAXX enhances phase separation of p62, or (b) DAXX is recruited to p62 bodies via the here identified protein/protein interaction and therefore increase the protein mass in the bodies which increases their volume fraction. The two scenarios have very different implications. To demonstrate enhancement of p62 phase separation by DAXX, the authors have to measure saturation concentrations of p62 in the presence and absence of DAXX. One way would be to do this via fluorescence intensities in the light phase of cells. Alternatively, they could reconstitute the system

in vitro. This would have the additional benefit of allowing them to directly quantify DAXX's influence on p62 oligomerization.

4. Fast recovery in FRAP measurements is neither necessary nor sufficient to show that a punctate structure has liquid droplet properties. The sentence "We confirmed that p62 underwent phase separation in HeLa cells by FRAP assays." is not warranted.

5. The authors say they want to test whether autophagy has an effect on DAXX-mediated regulation of p62 punctate formation. This is a surprising strategy, given that autophagy functions downstream of p62 body formation. They knock down autophagy effectors and then quantify the size/number of p62 bodies. These are not reduced compared to cells with normal autophagic function. However, I would have expected enhanced levels of p62 bodies if autophagy was inhibited because p62 bodies cannot be degraded by the autophagy machinery. There seem to be clues in their data (in Fig S3 and S4) that point towards this. This part of the manuscript requires at least substantial rewriting.

6. Showing that DAXX induces p62 oligomerization is not equivalent to showing that DAXX drives p62 phase separation. Many types of oligomerization that lead to oligomer species with defined size do not drive phase separation. Oligomerization may be able to enhance phase separation that is already happening via other interactions. A more quantitative approach of showing increased oligomerization of p62 in the presence of DAXX would be valuable and may help the authors disentangle oligomerization and phase separation. What types of oligomers does p62 form? Dimers, other discrete oligomers, or higher-order oligomers?

7. "Interestingly, we also observed more GFP-p62 breakdown products in the cells with both GFP-p62 and DAXX (Fig. 4b). This indicates that oligomeric p62 tends to be broken down by autophagy machinery." Just because DAXX enhances p62 oligomerization, and under these conditions they observe more p62 breakdown products, it is impossible to deduce that oligomeric p62 is broken down by the autophagy machinery. There are numerous other possibilities. Perhaps DAXX recruits ubiquitin ligases?

8. "Note that p62 levels in puromycin-treated cells were lower because puromycin treatment causes more p62 aggregation and in turn less soluble p62." This is a potential problem for the interpretation of the results. p62 and polyubiquitinated substrates may form solid aggregates via strong multivalent interactions, and these may effectively lead to autophagy. These aggregates should be analyzed under denaturing conditions.

9. There are several instances of overinterpretation in the discussion as well. "With the high-throughput Y2H screening, we characterise that DAXX interacts with p62 to facilitate p62 liquid phase separation and condensation, and DAXX confers p62 condensates more solid-like characters by increasing its condensation." This interpretation of the data is not warranted as laid out above in detail. "DAXX-induced p62 phase condensation is critical to facilitate p62 in recruitment of ubiquitinated proteins as well as its downstream factors." The authors do not show that DAXX is critical for the recruitment of ubiquitinated proteins via p62. Overall, the discussion is unfocused.

Minor comments:

1. What is the expected biological implication if DAXX indeed enhances p62 phase separation? DAXX is typically mostly nuclear. DAXX becomes at least partially cytoplasmic under oxidative conditions. Is this a consequence of its interaction with p62, or a prerequisite for the observed colocalization?
2. “Phase assembly” is not a typical naming convention. If the authors mean something different than dense phase or phase separation, they should specify it; otherwise it may make sense to use the more standard nomenclature.
3. P5: “The condensed p62 bodies are believed to more efficiently recruit aberrant proteins and specific factors in cytoplasm”. More efficient than what?
4. P6: “We further tested the direct p62-DAXX interaction by incubating bacterially expressed GST-p62 with 1-120aa deletion (GST-p62 Δ 120).” What does the deletion do and why was this construct chosen?
5. P7: “Interestingly, we observed that DAXX was recruited into p62 bodies to form DAXX bodies, and found that DAXX bodies always associated with the p62 bodies with stronger signals and bigger sizes in both HeLa (Fig. 1c) and HAP1 cells.” This sentence is unclear. Does DAXX form a new body, as the sentence suggests? Or is it recruited to p62 bodies? Which signals are stronger, and which sizes are bigger? Compared to what?
6. P11: “Given that DAXX induced p62 body formation, we investigated if DAXX drove p62 body phase condensation.” The authors do not show that DAXX induced p62 body formation (see my major comments 1 - 4). In addition, what is the difference between the two half sentences?
7. P11: “In both WT MEFs and ATG7 KO MEFs, DAXX significantly reduced the recovery levels after photobleaching (Fig. 3b-c), indicating that the role of DAXX in p62 droplet formation is independent of autophagy.” This conclusion cannot be reached from the data. DAXX/autophagy factors can have effects on p62 bodies that are different from changing FRAP properties.
8. P12: “we began to examine the interaction intensity between p62 molecules in the absence or the presence of DAXX.” The authors later refer to interaction affinity, and that is probably what they mean here, too.
9. P13: “Particularly, the effect of DAXX on GFP-p62 Δ PB1 oligomerisation appeared to be more phenomenal in the cells undergoing the treatment of puromycin.” A more descriptive adjective may be more appropriate.
10. P14: “Therefore, we conclude that DAXX effectively determines p62 oligomerisation.” DAXX is one factor that determines the degree of p62 oligomerization. p62 self-association and its concentration are other important factors.
11. Figure 6b(ii): The y-axis is labeled “Colocalization of p62 and Ub” and then values of several hundred are plotted. I would expect a percentage, which is clearly not what I am looking at. What are these values?
12. Figure 7d,e: The axis labels are missing in these panels. What are we looking at?

Dear Reviewers,

We are very grateful for your great efforts and constructive comments towards the manuscript. The revised manuscript and the detailed responses follow here. Please note that the revised text is highlighted in blue. Many thanks for your attentions.

Yours sincerely,

Shouqing Luo

Responses to the Reviewers' comments

Reviewer #1 (Remarks to the Author):

In this report, Yang et al. identify DAXX as a novel p62-interacting protein. DAXX mediates oligomerization of p62 in a PB1-independent manner, which drives p62 phase separation. DAXX–p62 condensates further recruit Keap1, leading to dysregulation of Nrf2 responses.

These results are novel and timely. However, as described in detail below, this study relies heavily on over-expression experiments and the role of endogenous DAXX is determined less robustly. Several different DAXX cell lines and RNAi-mediated knockdown were used, which makes this study rather complicated. Also, some in vitro studies would be required to conclude that DAXX indeed promotes phase separation.

Response: We are grateful for the Reviewer's critical comments. We have now comprehensively addressed these critical comments by considerable experiments. Please see the detailed responses below.

Major comments:

1. It could be true that DAXX interacts with p62 and promotes phase separation in vivo. However, these data do not necessarily indicate that DAXX itself promotes phase separation of p62. For example, ubiquitin or other proteins may mediate this process. To prove the authors' hypothesis, some in vitro studies are be required. Does purified DAXX actually promote p62 droplet formation in vitro? Without such data, the conclusion must be amended.

Response: We appreciate this insightful comment, and fully agree with the Reviewer on this important point. We have now purified the recombinant p62 and DAXX, and conducted the *in vitro* p62 phase separation assays as suggested (**Fig 4**). Our *in vitro* data show that DAXX plays a direct role in promoting p62 phase separation.

2. The role of endogenous DAXX in p62 droplet formation should be investigated more vigorously and consistently using knockout cells. The authors use HAP1 DAXX KO cells as a negative control in Figures 2 and 4 and DAXX KO MEFs in Figure 6. The results seem to be much clearer in KO MEFs than in KO HAP1 cells as there is only a small difference between WT and HAP1 DAXX KO cells in Figure 2c. The authors also use knockdown cells in some other experiments. It is highly recommended that the authors consistently use the same cell line (ideally KO MEFs) as a negative control throughout the study so that the experiments in different figures can be directly compared.

Response: We are thankful to the Reviewer for this point. We have now performed new experiments with DAXX KO MEFs, and new data (**Fig S4c**) consistently show that DAXX is critical in p62 puncta formation. Due to space limit, the data are kept as a supplementary figure.

We initially performed significant amount of experiments using DAXX KO HAP1 cells, as the cells were generated at the earlier stage of the project, and further confirmed our data with DAXX KO MEFs which were generated at a later time point. We hope to use different DAXX KO models to gain complementary data towards our conclusions.

3. The physiological importance of DAXX to formation of p62 bodies is not very clear. Do DAXX KO MEFs demonstrate smaller p62 droplets after treatment with puromycin (as in Figure 5), proteasome inhibitors, autophagy inhibitors (such as wortmannin), or arsenate?

Response: We have now added the data with puromycin treatment in DAXX WT/KO MEFs (in **Fig S4c**). The data consistently show that DAXX KO reduces p62 puncta formation after puromycin treatment. We thank the Reviewer for this helpful suggestion.

4. Figure 5a: The authors conclude that the PB1 domain is not required for oligomer formation of p62 by DAXX. However, the formation of condensates (Fig. 5a) and oligomers (Fig. 5b) of p62dPB1 induced by DAXX appears to be much milder than those of WT p62 (Figs. 1, 2, and 4). Rather, these results suggest that the PB1 domain is still important for DAXX-induced oligomer formation. It is also possible that the effect of PB1 deletion might have been overlooked missed because the PB1 deletion mutant possibly interacts with endogenous p62 through the UBA domain (PMID: 21715324). Thus, the p62dPB1 mutant should be introduced into p62 KO cells. Also, to test the role of PB1 domain, the K7R D69A mutant should be included.

Response: We agree with the Reviewer on this insightful point. As suggested by the Reviewer, we have now carried out the experiment using the p62 K7A D69A mutant (PB1-defective) in p62 KO MEFs for the effect of DAXX in the context. The data support that DAXX plays a PB1-independent role in p62 droplet formation (**Fig S9b**).

5. To determine the significance of p62-DAXX interaction, the authors use only deletion mutants in vivo. However, as the deleted regions are too large, it may affect intrinsic functions other than binding with p62. The authors should

narrow down the region required for the binding and use point mutants in the phase separation and Nrf2 activity experiments.

Response: We appreciate this critical point. We have now conducted site mutagenesis for p62 as well as DAXX. We have found that p62 P392L mutation reduces the p62-DAXX interaction. We have now performed the experiments with the p62 mutant for the role of DAXX in p62 puncta formation (**Fig S4b**). These data confirm the requirement of the p62-DAXX interaction for the role of DAXX in p62 puncta formation.

Likewise, DAXX D349A mutation also significantly reduced the interaction. With the mutant, we examined the effect of the DAXX mutant on Nrf2 activity with the luciferase assays (**Fig S11g-h**). These data further suggest the role of DAXX in Nrf2 activity via p62 phase separation. Please note that due to the space limitation, these important data are placed in the Supplementary figures.

We thank the Reviewer for these important points.

6. In Figure S3, why does the size of p62 puncta not increase in various Atg knockdown cells? This is contradictory to previously published results in many Atg knockout cells. The protein level of p62 should be checked in these knockdown cells to confirm that these Atg proteins are indeed depleted.

Response: We are grateful for the insightful point. Indeed, it is correct that p62 puncta are predicted to be bigger in autophagy-defective conditions. We observed that when cells were knocked down with Atg10 or Atg16L1 siRNA, overall p62-GFP exhibits bigger puncta, as shown in **Fig S6c (i)** (original **Fig S3c (i)**) (please see, e.g., **Column 1; 7; 10** in **Fig S6c(i)**, for convenience), where the size of all p62 puncta was quantified. However, we unexpectedly observed that, in the case of Atg5 knockdown, the size of p62 puncta was markedly reduced (see **Column 1** and **4** in **Fig. S6c (i-iii)**). Although such a phenomenon is not well understood, we reason that Atg5 could play a direct role in p62 puncta formation.

We have previously validated the knockdown efficiency for the Atg proteins in all cases where knockdown experiments were performed. In this case, **Fig S7b** (original **Fig S4b**) confirmed the knockdown efficiency.

We have now added a paragraph to discuss the size of p62 puncta in **p12**:

"Notably, when p62-GFP HeLa Tet-on stable cells were transfected with control siRNA, Atg10 or Atg16L1 siRNA, along with vector control (without DAXX over-expression), overall p62-GFP exhibited bigger puncta in the cells with knockdown of

Atg10 or Atg16L1, compared to those in control knockdown cells (see Column 1, 7 and 10 in **Fig. S6c (i)**), where the size of all p62 puncta was quantified. These suggest that p62 puncta size may be modulated by autophagy. However, the effect of autophagy on p62 puncta size was no longer observed in the cells with DAXX over-expression (see Column 2 and 3, 8 and 9, 11 and 12 in **Fig. S6c (i)**). This suggests that the effect of DAXX on p62 puncta size could mask that of defects in autophagy. Interestingly, we unexpectedly observed that, in the case of Atg5 knockdown, the size of p62 puncta was always markedly reduced (see Column 1 and 4 in **Fig. S6c (i-iii)**). Although this phenomenon is currently not well understood, we reason that ATG5 could directly play a role in p62 puncta formation." (**para 2, p12**).

7. Figure 3: The data suggest that DAXX induces more gel-like structures rather than liquid droplets. This should be mentioned.

Response: We agree with the Reviewer on this point. We have now added this in the text (**line 5-6, p13**).

8. The mRNA and protein levels of p62 should be determined in DAXX KO and DAXX overexpressed cells to confirm that the effect does not result from transcriptional alteration.

Response: This is indeed an important point. We have now included these data of p62 mRNA expression in DAXX-overexpressed, or DAXX knockdown cells (**Fig S5e-f**). Our data show that DAXX does not increase p62 mRNA levels. We also show that DAXX does not affect p62 at the protein level in DAXX knockdown or overexpression cells (**Fig S5c-d**). We thank the Reviewer for this critical comment.

9. Fig. 1d needs a negative control.

Response: The negative controls in **Fig 1g** (original **Fig 1d**) has now been included.

Minor comments:

1. In Introduction or the first paragraph of Result, it should be mentioned that previous studies suggest that ubiquitin can promote phase separation or droplet formation of p62 (Sun et al. Cell Res 2018 (ref #31), Zaffagnini et al. EMBO J. 2018).

Response: We have now added the text in the first para of the Result (**the last 3 lines, p5; the first 2 lines, p6**), and discussed these in the Discussion (**line 4-7, p22**). We are thankful to the Reviewer for this pertinent point.

2. Figure 2: The expression level of p62 and DAXX should also be determined by immunoblotting. Ideally, the Tet system should be used to express DAXX instead of p62 if the authors want to see the effect of DAXX on p62 droplet formation.

Response: The expression levels of p62 or DAXX by immunoblotting have now been included (**Fig 2**).

We have now newly generated stably-expressing DAXX Tet-on expression cell lines, and tested the experiments as suggested by the Reviewer for the role of DAXX in p62 droplet formation (**Fig S4a**). Our data confirm the effect of DAXX on p62 puncta formation in the cells.

We thank the Reviewer for these constructive points.

3. Figure 4b: To prove that the GFP fragment is a result of autophagic degradation, ATG knockout or knockdown cells should be used.

Response: Komatsu and colleagues have previously shown that the levels of free GFP positively correlate with autophagic degradation rate of GFP-p62 in cells (PMID: 24492006). We have now utilised Beclin 1 knockdown, and confirm that the free GFP is a result of GFP-p62 autophagic degradation (**Fig. S9a**). We have now added these in the text (**the end of para 1, p15**).

We are very grateful for the Reviewer's insightful and constructive comments, which have been very helpful throughout this revision.

Reviewer #2 (Remarks to the Author):

This manuscript builds on the previous extensive studies of the authors on the role and regulation of autophagy. In this study, however, they have identified the role of cytoplasmic DAXX in driving p62 liquid phase condensation by inducing its oligomerisation. Furthermore, they show the role of DAXX mediated p62 phase condensation as a mechanism of stress adaptation acting via the Nrf2 pathway.

Recent studies emphasized the concept that stress adaptation through protein phase separation or open macromolecular assemblies is a widespread phenomenon in nature. Phase separating proteins including p62 are associated with age related diseases and phase transition is proposed to drive progression of such diseases. In this context the current manuscript provides interesting insights into this biological phenomena.

However the authors should discuss and address some critical concerns:

1. The study is based mostly on two cell lines. Considering the conflicting literature on Daxx localization (nuclear vs cytoplasmic), it is critical to assess the proposed interactions between Daxx and p62 in more physiological systems and ideally also in tissues (at least by immunofluorescence). In this respect, access to tumor tissues displaying loss of Daxx expression (LoF mutations are found in a number of human neoplasms) would allow the authors to correlate loss of Daxx to alterations of p62 distribution in disease-relevant settings. Finally, it is surprising that the authors have not discussed how interaction between Daxx and a p62/Nrf2 pathway might affect Daxx H3.3 chaperone activity.

Response: We are grateful for the Reviewer's critical comments. We agree that it is critical to test the p62-DAXX interaction in physiological conditions. We have now added the immunoprecipitation data (**Fig 1c**), and the data of p62-DAXX colocalisation (by immunofluorescence) using mouse striatal tissues (**Fig 1e**). We trust that our data with DAXX conditional knockout mice and other DAXX ablation models should address DAXX LoF in p62 puncta formation.

We are thankful to the Reviewer for the insightful comment on the discussion about DAXX H3.3 chaperone activity. We have now included the discussion (**line 4-7, para 2, p23**).

2. Foremost, the p62 deficiency has been proposed to enhance proteasome activity, which degrades polyubiquitinated proteins (Korolchuk et al., 2009;

Moscat and Diaz-Meco, 2009), hence it critical to evaluate the effect of the DAXX mediated p62 liquid phase condensation on the half life p62 itself and on the proteosomal activity and polyubiquitinated proteins levels over time in these cells.

Response: We thank the Reviewer for the important points. We have now tested the proteasomal activity in DAXX knockdown or DAXX overexpression cells, accordingly. We did not observe a significant change in proteasomal activity in DAXX knockdown or overexpression cells (**Fig S5a-b**).

We have also tested the half-life of p62 in DAXX knockdown or overexpression cells using cycloheximide (CHX) treatment in a time course. The p62 levels were not subjected to significant changes in either DAXX knockdown or overexpression cells (**Fig S5c-d**).

3. To further increase the impact of this study, authors should look into the effect of P62 mutations has been identified in Paget´s disease of the Bone and ALS disease on Daxx interaction and its ability to affect p62 homeostasis. While most such mutations lie in the polyubiquitin chain binding UBA domain and not in the PEST domain identified by the authors as the P62 domain interacting with DAXX, its is pertinent to evaluate if DAXX mediated p62 body formation is altered by such mutations and this could add important details to the as of yet unclear mechanism behind these mutation related pathologies.

Response: We appreciate this interesting point. We have now investigated the effect of the disease-relevant p62 mutations on DAXX interaction (**Fig S1e**). We have found that p62 P392L mutation reduces the p62-DAXX interaction - this suggests that this region also contributes to the p62-DAXX interaction. We have now performed the experiments with the p62 mutant for the role of DAXX in p62 puncta formation (**Fig S4b**). Our data confirm the requirement of the p62-DAXX interaction for the role of DAXX in p62 puncta formation. Due to the space limitation, these important data are placed in the Supplementary figures. We thank the Reviewer for this critical suggestion.

4. While all p62 puncta colocalizes with DAXX puncta, not all DAXX puncta are positive for p62, the authors to discuss this interesting disparity.

Response: We appreciate the Reviewer's insightful observation. Such a colocalisation disparity may suggest that DAXX would be required for efficient p62 phase separation, and DAXX might also play p62-independent roles in cytoplasm. We have now discussed this (**line 5-7, para 2, p7**). We thank the Reviewer for the interesting point.

Minor comments:

1. The haploid HAP1 cells has been shown to have elevated levels of p53, a known DAXX interacting protein and might influence the assays with DAXX. Hence it would be important to perform the KO assays in diploid cells.

Response: Yes, this is an important point. We used DAXX KO/KD assays in a variety of diploid cells, e.g. DAXX KO MEFs, DAXX KD HeLa, to complement and corroborate the data with HAP1 cells.

2. It is difficult to appreciate the Fig.2 c claiming that “The formation of p62 puncta is weakened in DAXX knockout HAP1 cells”, as the puncta like structures are present in the DAXX KO HAP1 cells as well.

Response: Thanks for this point. We have now amended the sentence to: “The formation of p62 puncta is reduced in DAXX knockout HAP1 cells” (line 1, para 2, p42). We have now added the data with DAXX KO MEFs (Fig S4c) to complement the data of HAP1 cells.

3. The quality of the gel image makes it impossible to appreciate the Fig. 6, d, claiming “p62-polyubiquitinated proteins interaction is weakened in DAXX-KO HAP1 cells” and the image quality is as such below the Journal publication standards.

Response: We thank the Reviewer for this pertinent point. Please note the relatively low levels of endogenous polyubiquitin proteins in the cells. In this WB (now Fig 7d), we used the P4D1 ubiquitin antibody (which would be one of the best ubiquitin antibodies available).

4. Quality of WB data in Figure 3e, 4b and 5c is below standard and makes conclusions related to these data not very convincing.

Response: We are thankful to the Reviewer for this relevant point. In Fig 3e, the samples were treated with urea, which causes smearing effects. In Fig 4b (now Fig 5b), please note that native PAGE conditions usually result in the effects. For Fig 5c (now Fig 6c), depending on the oligomerisation nature of a protein, protein oligomerisation could cause smearing effects. We found that this had always been the case for GFP-p62 del PB1 protein.

5. It would be important to provide details on the efficiency and level of DAXX KD in the UAS-DAXX shRNA drosophila experiment to evaluate the correlation with decrease in ubiquitin puncta as proposed in the Fig 6, c.

Response: Yes, we have now performed the qPCR experiments to measure DAXX/DLP knockdown efficiency. Please see the new data (**Fig S10d**).

6. *The Thymus has been misspelled twice in the Author Contributions (page 30) and Fig 2, d. legend.*

Response: We have now corrected the misspellings. We thank the Reviewer for the corrections.

We are very thankful to the Reviewer for the constructive and thoughtful efforts towards our manuscript.

Reviewer #3 (Remarks to the Author):

The manuscript by Yang et al identifies the protein DAXX as a binding partner of p62. The authors aim to test the effect of DAXX on p62 phase separation and function. While they generate interesting data, their interpretation that p62 undergoes phase separation and DAXX enhances it, is not justified. While the insight that DAXX is a player in the p62/Keap/Nrf2 signaling axis is interesting and important, the work is overall too preliminary for publication in this reviewer's opinion.

Response: We are thankful to the Reviewer's critiques. We have now significantly amended the manuscript with the critical experiments as suggested by the Reviewer. We hope the revised manuscript is now satisfactory to the Reviewer.

My major comments are as follows:

1. The authors do not demonstrate that the system they study actually undergoes liquid-liquid phase separation. They show punctate structures in the cytoplasm of cells. If we consider that p62 phase separation has recently been demonstrated by Su et al. (ref 31 in this manuscript), we would still need to see evidence that DAXX enhances phase separation of p62. We only see more/larger punctate structures in the presence of DAXX.

Response: We have now added the data of *in vitro* p62 phase separation using recombinant p62 and DAXX proteins (**Fig 4a-e**). These data clearly show that DAXX directly promotes p62 phase separation. We thank the Reviewer for this critical point.

2. To compare different conditions, the authors measure the size of the largest punctum in cells and the percentage of cells with particularly large puncta (> 1 um) as a surrogate for the volume fraction of the "dense phase" (i.e. the p6 bodies). It would be far superior to measure the actual volume fraction (i.e. the total volume of bodies compared to the total volume of cells). If the authors do not do this, they should at least discuss that their measure is a suboptimal surrogate of what they should really be measuring.

Response: We are thankful to the Reviewer for this suggestion. In this work, we initially found that DAXX promotes p62 body size, and we aimed to focus on this consistently. In our conditions, it was technically challenging for us to accurately measure the cellular volumes due to the complexity in the shapes of the cells used in this work. We have now added the discussion regarding this, as suggested by the Reviewer (**para 2, p21**).

3. Possible explanations for observing a larger volume fraction of p62 bodies are the following: (a) DAXX enhances phase separation of p62, or (b) DAXX is recruited to p62 bodies via the here identified protein/protein interaction and therefore increase the protein mass in the bodies which increases their volume fraction. The two scenarios have very different implications. To demonstrate enhancement of p62 phase separation by DAXX, the authors have to measure saturation concentrations of p62 in the presence and absence of DAXX. One way would be to do this via fluorescence intensities in the light phase of cells. Alternatively, they could reconstitute the system *in vitro*. This would have the additional benefit of allowing them to directly quantify DAXX's influence on p62 oligomerization.

Response: We appreciate the Reviewer's insightful thoughts. We have now demonstrated that DAXX directly increases p62 phase separation using *in vitro* reconstitution system (Fig 4).

4. Fast recovery in FRAP measurements is neither necessary nor sufficient to show that a punctate structure has liquid droplet properties. The sentence "We confirmed that p62 underwent phase separation in HeLa cells by FRAP assays." is not warranted.

Response: We have now amended this to: "We first confirmed that p62-GFP fluorescence signal could undergo recovery after photobleaching in HeLa cells" (the last 2 lines, p12).

5. The authors say they want to test whether autophagy has an effect on DAXX-mediate regulation of p62 punctate formation. This is a surprising strategy, given that autophagy functions downstream of p62 body formation. They knock down autophagy effectors and then quantify the size/number of p62 bodies. These are not reduced compared to cells with normal autophagic function. However, I would have expected enhanced levels of p62 bodies if autophagy was inhibited because p62 bodies cannot be degraded by the autophagy machinery. There seem to be clues in their data (in Fig S3 and S4) that point towards this. This part of the manuscript requires at least substantial rewriting.

Response: We appreciate the Reviewer's critical comment. We have previously addressed this in the response to the Reviewer 1's Comment 6, we are glad to have this opportunity to further discuss this.

The experiments were tested, since autophagy would affect p62 puncta size, as the Reviewer correctly pointed out here. We observed that when cells were knocked down with Atg10 or Atg16L1 siRNA, overall p62-GFP exhibits bigger puncta, as shown in **Fig S6c (i)** (original **Fig S3c (i)**) (please see, e.g., **Column 1; 7; 10** in **Fig S6c(i)**, for convenience), where the size of all p62 puncta was quantified. However, we unexpectedly observed that, in the case of Atg5 knockdown, the size of p62 puncta was reduced (please see **Column 1 and 4** in **Fig. S6c (i-iii)**). Although such a phenomenon is not well understood, we reason that Atg5 could play a direct role in p62 puncta formation.

We have now substantially amended this part by adding a paragraph of discussion (**para 2, p12**):

"Notably, when p62-GFP HeLa Tet-on stable cells were transfected with control siRNA, Atg10 or Atg16L1 siRNA, along with vector control (without DAXX over-expression), overall p62-GFP exhibited bigger puncta in the cells with knockdown of Atg10 or Atg16L1, compared to those in control knockdown cells (see **Column 1, 7 and 10** in **Fig. S6c (i)**), where the size of all p62 puncta was quantified. These suggest that p62 puncta size may be modulated by autophagy. However, the effect of autophagy on p62 puncta size was no longer observed in the cells with DAXX over-expression (see **Column 2 and 3, 8 and 9, 11 and 12** in **Fig. S6c (i)**). This suggests that the effect of DAXX on p62 puncta size could mask that of defects in autophagy. Interestingly, we unexpectedly observed that, in the case of Atg5 knockdown, the size of p62 puncta was always markedly reduced (see **Column 1 and 4** in **Fig. S6c (i-iii)**). Although this phenomenon is currently not well understood, we reason that ATG5 could directly play a role in p62 puncta formation."

6. Showing that DAXX induces p62 oligomerization is not equivalent to showing that DAXX drives p62 phase separation. Many types of oligomerization that lead to oligomer species with defined size do not drive phase separation. Oligomerization may be able to enhance phase separation that is already happening via other interactions. A more quantitative approach of showing increased oligomerization of p62 in the presence of DAXX would be valuable and may help the authors disentangle oligomerization and phase separation. What types of oligomers does p62 form? Dimers, other discrete oligomers, or higher-order oligomers?

Response: We appreciate this critical point, and agree that p62 oligomerisation is not identical to p62 phase separation.

Here we aimed to provide mechanistic insight into the role of DAXX in enhancing p62 phase separation/condensation. p62 alone is known to undergo basal levels of

phase separation, and DAXX promotion of p62 oligomerisation would mechanistically lead to enhanced p62 phase separation. We have now added a discussion towards this (**from 6th last line, p22**): "The regulation of protein oligomerisation would be one of the key factors for protein phase separation. Previous reports (ref 32, 44) as well as our current data suggest that p62 alone undergoes basal phase separation. We show here that DAXX regulates p62 oligomerisation and promotes p62 phase separation."

According to our capillary-based electrophoresis (Wes size) assays, p62 can exist in monomers, dimers, trimers, tetramers and higher-order oligomers.

7. *“Interestingly, we also observed more GFP-p62 breakdown products in the cells with both GFP-p62 and DAXX (Fig. 4b). This indicates that oligomeric p62 tends to be broken down by autophagy machinery.” Just because DAXX enhances p62 oligomerization, and under these conditions they observe more p62 breakdown products, it is impossible to deduce that oligomeric p62 is broken down by the autophagy machinery. There are numerous other possibilities. Perhaps DAXX recruits ubiquitin ligases?*

Response: We are thankful to the Reviewer for this pertinent point. Komatsu and colleagues have previously shown that the levels of free GFP positively correlate with autophagic degradation rate of GFP-p62 in cells (PMID: 24492006). Our new experiment shows that free GFP was reduced in autophagy-defective conditions, confirming that free GFP is a result of GFP-p62 autophagic degradation (**Fig S9a**). We have now added these in the text (**the end of para 1, p15**).

8. *“Note that p62 levels in puromycin-treated cells were lower because puromycin treatment causes more p62 aggregation and in turn less soluble p62.” This is a potential problem for the interpretation of the results. p62 and polyubiquitinated substrates may form solid aggregates via strong multivalent interactions, and these may effectively lead to autophagy. These aggregates should be analyzed under denaturing conditions.*

Response: We thank the Reviewer for this consideration. In this experiment, 1) the analysis was performed under SDS-PAGE denaturing conditions; 2) in our data, we aimed to detect the interaction between soluble p62 and ubiquitinated proteins in the presence/absence of DAXX, by immunoprecipitation. Technically, only soluble proteins can be used for immunoprecipitation; 3) the levels of the physical interaction between p62 and ubiquitinated proteins were complemented with immunostaining (**Fig 7a-b; Fig S10 a-c**).

9. There are several instances of overinterpretation in the discussion as well. “With the high-throughput Y2H screening, we characterise that DAXX interacts with p62 to facilitate p62 liquid phase separation and condensation, and DAXX confers p62 condensates more solid- like characters by increasing its condensation.” This interpretation of the data is not warranted as laid out above in detail. “DAXX-induced p62 phase condensation is critical to facilitate p62 in recruitment of ubiquitinated proteins as well as its downstream factors.” The authors do not show that DAXX is critical for the recruitment of ubiquitinated proteins via p62. Overall, the discussion is unfocused.

Response: We have now significantly rewritten the discussion (p21-24). We hope, with the new data and the amended discussion, these are now improved (please note that **Fig S10** (original **Fig S6**) shows the data regarding the role of DAXX in recruitment of ubiquitinated proteins by p62). We are grateful to the Reviewer for the constructive comments and suggestions.

Minor comments:

1. What is the expected biological implication if DAXX indeed enhances p62 phase separation?

Response: p62 body formation has multiple established and potential functions: recruitment of proteins for selective autophagic clearance; recruitment of proteins for storage; recruitment of KEAP1; autophagosome formation; autophagosome membrane bending. Our work suggests that the role of DAXX in p62 phase separation/condensation would be important for the recruitment of ubiquitinated protein for autophagic clearance, and KEAP1 recruitment, playing a role in Nrf2 pathways.

DAXX is typically mostly nuclear. DAXX becomes at least partially cytoplasmic under oxidative conditions. Is this a consequence of its interaction with p62, or a prerequisite for the observed colocalization?

Response: DAXX was suggested to have signals for both nuclear localisation and export (PMID:15128734; PMID: 17661348). At present it is not clear if p62 is critical for the cytoplasmic localisation of DAXX.

2. “Phase assembly” is not a typical naming convention. If the authors mean something different than dense phase or phase separation, they should specify it; otherwise it may make sense to use the more standard nomenclature.

Response: We appreciate this point. We have now amended this into: “Phase separation”.

3. P5: “The condensed p62 bodies are believed to more efficiently recruit aberrant proteins and specific factors in cytoplasm”. More efficient than what?

Response: We have now corrected the sentence into: “The condensed p62 bodies are believed to efficiently recruit aberrant proteins and specific factors in cytoplasm.” (line 4-5, para 2, p5).

4. P6: “We further tested the direct p62-DAXX interaction by incubating bacterially expressed GST-p62 with 1-120aa deletion (GST-p62 Δ 120).” What does the deletion do and why was this construct chosen?

Response: p62 1-120aa represents the PB1 domain, which is known to be required for p62 oligomerisation. We chose this deletion as we initially aimed to test if the PB1 domain is needed for the interaction.

5. P7: “Interestingly, we observed that DAXX was recruited into p62 bodies to form DAXX bodies, and found that DAXX bodies always associated with the p62 bodies with stronger signals and bigger sizes in both HeLa (Fig. 1c) and HAP1 cells.” Thi sentence is unclear. Does DAXX form a new body, as the sentence suggests? Or is it recruited to p62 bodies? Which signals are stronger, and which sizes are bigger? Compared to what?

Response: We have now amended the sentence: “Interestingly, we observed that DAXX was recruited into p62 bodies, and found that p62 bodies with DAXX signals always exhibited stronger signals and bigger sizes (**Fig. 1d-f**), compared to those without DAXX signals.” (line 2-5, para 2, p7). Hope this is now clearer. We thank the Reviewer for the critical thinking.

6. P11: “Given that DAXX induced p62 body formation, we investigated if DAXX drove p62 body phase condensation.” The authors do not show that DAXX induced p62 body formation (see my major comments 1 - 4). In addition, what is the difference between the two half sentences?

Response: We have now amended the sentence into “Given that DAXX promoted p62 puncta formation.....” (line 2, p13). We have now added new data (e.g. **Fig 4**) to show that DAXX directly enhances p62 phase separation *in vitro*. We thank the Reviewer for the pertinent point.

7. P11: *“In both WT MEFs and ATG7 KO MEFs, DAXX significantly reduced the recovery levels after photobleaching (Fig. 3b-c), indicating that the role of DAXX in p62 droplet formation is independent of autophagy.” This conclusion cannot be reached from the data. DAXX/autophagy factors can have effects on p62 bodies that are different from changing FRAP properties.*

Response: We appreciate this point. We have now changed the sentence into “..... indicating that the role of DAXX in the FRAP property of p62 droplet is independent of autophagy” (line 9-10, p13).

8. P12: *“we began to examine the interaction intensity between p62 molecules in the absence or the presence of DAXX.” The authors later refer to interaction affinity, and that is probably what they mean here, too.*

Response: We have now replaced “interaction intensity” with “interaction affinity” (line 3, para 2, p14).

9. P13: *“Particularly, the effect of DAXX on GFP-p62 Δ PB1 oligomerisation appeared to be more phenomenal in the cells undergoing the treatment of puromycin.” A more descriptive adjective may be more appropriate.*

Response: Yes, we amended the “phenomenal” into “pronounced” (the 3rd last line, p16).

10. P14: *“Therefore, we conclude that DAXX effectively determines p62 oligomerisation.” DAXX is one factor that determines the degree of p62 oligomerization. p62 self-association and its concentration are other important factors.*

Response: We have now corrected this with the new sentence: “Therefore, we conclude that DAXX is a factor to determine the degree of p62 oligomerisation” (the last line, para 1, p17). We thank the Reviewer for the insightful thought.

11. Figure 6b(ii): *The y-axis is labeled “Colocalization of p62 and Ub” and then values of several hundred are plotted. I would expect a percentage, which is clearly not what I am looking at. What are these values?*

Response: The values of Y-axis, representing the area of colocalisation, were acquired from the quantification programme (LAS AF Lite). We have now normalised the values by setting the mean of the controls as 100 (now **Fig 7a (ii), b(ii)**). We thank the Reviewer for this pertinent point.

12. Figure 7d,e: The axis labels are missing in these panels. What are we looking at?

Response: We thank the Reviewer for this helpful point. We have now added the labels (Y-axis for cell counts and X-axis for FITC levels) (now **Fig 8d-e**).

We are very thankful to the Reviewer for these critical and constructive comments.

Reviewers' comments:

Reviewer #1 (Remarks to the Author):

Yang et al. have revised this manuscript with some additional data, but the manuscript continues to have critical issues. In particular, the direct evidence that DAXX induces phase separation of p62 is still weak. The new in vitro data in Fig. 4 are not convincing and could be even contradictory to the authors' hypothesis. Listed below are specific comments.

(The numbers correspond to those of the previous major comments.)

1. Now the authors provide a set of in vitro data, but the results are not convincing. The structures shown in new Fig. 4 do not look like typical liquid droplets. Although the authors state in the text that these are "spherical", but actually they do not appear spherical. Typically, phase separated droplets are truly spherical and often fuse with each other. It is more important to show that these structures are "dynamic". FRAP analysis would be essential. Overall, the present data are not enough to conclude that DAXX induces phase separation of p62. These in vitro findings rather suggest that the effect of DAXX in vivo may be secondary.

4. The authors have tested the role of PB1 in p62 KO cells (Fig. S9b). However, since an essential positive control (full-length wild-type p62) is missing in this experiment, no one can tell whether the PB1 mutant demonstrates full activity. It seems to be less efficient. Thus, it is still possible that the PB1 domain is important even though it is not absolutely required. Also related to Comment #1, an in vitro experiment using the PB1 mutants would be most suitable to address this point.

6. The response to previous Comment #6 is not sufficient. It is important to determine whether knockdown of these Atg genes are efficient and the p62 protein accumulates. As both Atg10 and Atg5 are required for LC3 lipidation, immunoblotting of LC3 would be the best method to confirm a block in autophagic flux. Also, immunoblotting of p62 is essential.

Reviewer #2 (Remarks to the Author):

The authors have addressed most of my comments. As a result, the manuscript is definitely improved. I don't have further comments

Reviewer #3 (Remarks to the Author):

The authors have addressed my comments to my satisfaction.

(Editor's note - Reviewer #3 offered additional feedback after the Decision was sent):
Additional Reviewer #3's comments on Reviewer #1's latest report:

I disagree with the idea that the droplets have to be liquid to be formed by liquid-liquid phase separation. The proteins may encode viscoelastic material properties and there may be fast aging of the droplets into gel-like structures. However, I agree that the microscopy images are of poor quality and make it impossible to judge whether these are really liquid droplets or some other amorphous or even fibrillar aggregates. It is possible to interrogate the system for LLPS features by changing the input concentration into a phase separation assay and monitoring whether the concentrations in the light and dense phase remain constant. This would be a prudent way forward.

Reviewer 1#

Yang et al. have revised this manuscript with some additional data, but the manuscript continues to have critical issues. In particular, the direct evidence that DAXX induces phase separation of p62 is still weak. The new in vitro data in Fig. 4 are not convincing and could be even contradictory to the authors' hypothesis. Listed below are specific comments.

(The numbers correspond to those of the previous major comments.)

1. Now the authors provide a set of in vitro data, but the results are not convincing. The structures shown in new Fig. 4 do not look like typical liquid droplets. Although the authors state in the text that these are "spherical", but actually they do not appear spherical. Typically, phase separated droplets are truly spherical and often fuse with each other. It is more important to show that these structures are "dynamic". FRAP analysis would be essential. Overall, the present data are not enough to conclude that DAXX induces phase separation of p62. These in vitro findings rather suggest that the effect of DAXX in vivo may be secondary.

Response: Phase-separated **liquid** droplets are expected to have a spherical shape. The perfect-sphere (truly spherical) shape should only apply to liquid-like droplets. The droplet shape in reality can be complex (see below), and depends on experimental settings (e.g. protein property, salt conditions, surface conditions, imaging conditions, incubation time).

When liquid-like droplets undergo transitions to form viscous/gel-like or solid-like assemblies, they can have deformed spherical or irregular structures (example ref: Alberti et al, Cell, 2019, PMID: 30682370). For example, once the liquid droplets of Tau become more solid-like structures, they lose their sphere-like shape (Wegmann et al, EMBO J, 2018, PMID: 29472250). The co-existence of intermediate states between liquid-like, gel-like and solid-like phase would result in a complexity in protein droplet shape.

More specifically, p62 droplets have been suggested to be viscous structures by Yu and colleagues (Sun et al, Cell Res, 2018, PMID: 29507397). In our manuscript (**p13**), we have also proposed that p62 droplets, particularly with DAXX, exhibit viscous/gel-like properties (as mentioned in **Reviewer 1's previous Comment 7**). Therefore, we expect that p62 droplets/DAXX may have deformed spherical structures (deformed spherical structures for gel-like droplets, see the 2nd last para, p427, the review article by Alberti et al, Cell, 2019, PMID: 30682370). Yu and colleagues first showed the shapes of p62 droplets (Sun et al, Cell Res, 2018, PMID: 29507397). Please see exemplar images excerpted from the paper - the *in vitro* p62 droplets are not spherical (**Appendix 1**). To further clarify this, we have added discussions about the shape of p62 droplets in the revised manuscript (**p14**).

We have now improved the image quality with a new imaging approach (**Fig 4**).

Moreover, our sedimentation experiments convincingly suggest that DAXX promotes p62 droplet formation.

Regarding FRAP experiments, we used droplet fusion to demonstrate p62 droplet wetting property (non-fluorescent recombinant p62 was used here) - this is a standard approach to show the droplet fluidity (Shin and Brangwynee, Science, 2017, PMID: 28935776). FRAP is an alternative assay method to show droplet fluidity for fluorescent proteins. In fact, no FRAP (or *in vitro* FRAP) experiments were carried out in many phase separation papers (examples: Larson et al, Nature, 2017, PMID: 28636604; Case et al, Science, 2019, PMID: 30846599; Yoshizawa et al, Cell, 2018, PMID: 29677513). We consider, for *in vitro* assays, the p62 droplet fusion assay could be a better approach, because it does not require a fusion fluorescent tag, which might influence the physical properties of the protein. In addition, we have included FRAP data for cellular p62 (**Fig 3**).

4. The authors have tested the role of PB1 in p62 KO cells (Fig. S9b). However, since an essential positive control (full-length wild-type p62) is missing in this experiment, no one can tell whether the PB1 mutant demonstrates full activity. It seems to be less efficient. Thus, it is still possible that the PB1 domain is important even though it is not absolutely required. Also related to Comment #1, an *in vitro* experiment using the PB1 mutants would be most suitable to address this point.

Response: PB1-mutated p62 ineffectively forms puncta in cells. In basal conditions, compared to WT-p62, PB1-mutated p62 form much smaller and fewer puncta, even in the presence of DAXX. For *in vitro* assays, it is very unlikely that PB1-mutated p62 effectively forms droplets *in vitro* (even in the presence of DAXX) - please note that protein local concentration *in vitro* is much lower than that in cells. Therefore, *in vitro* PB1-mutated p62 experiment unlikely yields meaningful data.

For this part of the work, we have shown that: **1.** DAXX can interact with p62 in the absence of PB1 domain; **2.** DAXX promotes p62 Δ PB1 puncta formation in cells (especially with puromycin); **3.** DAXX promotes p62 Δ PB1 oligomerisation *in vitro*; **4.** DAXX promotes p62 Δ PB1 oligomerisation in cells; **5.** DAXX promotes PB1-mutated p62 puncta formation.

We have now added the data showing the effect of DAXX on WT-p62 puncta formation in p62 KO MEFs (**Fig S9b**). These data further suggest that the role of DAXX in p62 puncta formation is independent of PB1 domain.

6. The response to previous Comment #6 is not sufficient. It is important to determine whether knockdown of these Atg genes are efficient and the p62 protein accumulates. As both Atg10 and Atg5 are required for LC3 lipidation, immunoblotting of LC3 would be the best method to confirm a block in autophagic flux. Also, immunoblotting of p62 is essential.

Response: Previously we have shown that the proteins are largely knocked down by siRNAs of Atg5, Atg10 or Atg16L1 by immunoblots (**Fig S7b**) - this is a standard assay for the confirmation of protein knockdown.

We have now confirmed that knockdown of Atg5, Atg10 or Atg16L1 reduced LC3-II levels and increased p62 levels. It is well known that Atg5, Atg10 or Atg16L1 is required for autophagosome formation (marked by LC3-II levels), and autophagy (marked by p62 clearance). Please see the data in **Appendix 2**.

Reviewer 3#

I disagree with the idea that the droplets have to be liquid to be formed by liquid-liquid phase separation. The proteins may encode viscoelastic material properties and there may be fast aging of the droplets into gel-like structures. However, I agree that the microscopy images are of poor quality and make it impossible to judge whether these are really liquid droplets or some other amorphous or even fibrillar aggregates.

It is possible to interrogate the system for LLPS features by changing the input concentration into a phase separation assay and monitoring whether the concentrations in the light and dense phase remain constant. This would be a prudent way forward.

Response: We appreciate that Reviewer 3 confirms that the criticisms listed by Reviewer 1 can be addressed. We have now improved the image quality with a new imaging approach (**Fig 4**), as suggested.

For the LLPS assays, we have always excluded any potential aggregates by centrifuging the protein solutions immediately before the droplet formation assays.

We are grateful to the Reviewers for the constructive comments.

Appendix 1

In vitro p62 droplets from Sun et al, Cell Research, 28(4):405-415, 2018, PMID: 29507397

Fig 2h

Fig 2l

Fig 6f

Appendix 2

Appendix-2. Knockdown of autophagy machinery genes reduces LC3-II levels and increases p62 levels in p62-GFP expressing HeLa Tet-on cells

HeLa Tet-on cells harbouring p62-GFP were transfected with control siRNA, Atg5 siRNA or Atg16L1 siRNA, respectively. After 24 hours, the cells were induced with Doxycycline to express p62-GFP for 24 hours. Cells were collected for immunoblot to confirm the knockdown efficiency. **(a)** anti-Atg5, p62, LC3 or b-actin was used for the immunoblotting; **(b)** anti-Atg10, p62, LC3 or b-actin was used for the immunoblotting; **(c)** anti-Atg16L1, p62, LC3 or b-actin was used for the immunoblotting. Note that the LC3 antibody predominantly detects LC3-II in HeLa cells.

endo-p62: endogenous p62; Lighter: lighter (shorter) exposure; Heavier: heavier (longer) exposure.

REVIEWERS' COMMENTS:

Reviewer #1 (Remarks to the Author):

The authors have addressed most of my previous concerns. However, I am still not convinced by the authors' argument that the observed structures are not spherical because liquid-to-gel transition has occurred. However, the experiments in Fig. 4b are done in vitro. To avoid the liquid-to-gel transition and see the initial structures, the authors can simply shorten the incubation time. I do not understand why this is not feasible. Nonetheless, the images in new Fig. 4 (at 1-h incubation) has been improved and may be published as images after a long-term (1 h) incubation (although I still think that showing the structures after short-term incubation and a FRAP analysis would strengthen the conclusion).

The authors have now shown the effects of ATG gene knockdown on p62 accumulation and LC3-II formation as a figure for reviewers. However, the differences are not large, and LC3-I does not accumulate. I suggest that this data should be published in supplemental figures so that readers can judge how much the knockdown of these proteins is effective.

REVIEWERS' COMMENTS:

Reviewer #1 (Remarks to the Author):

The authors have addressed most of my previous concerns. However, I am still not convinced by the authors' argument that the observed structures are not spherical because liquid-to-gel transition has occurred. However, the experiments in Fig. 4b are done in vitro. To avoid the liquid-to-gel transition and see the initial structures, the authors can simply shorten the incubation time. I do not understand why this is not feasible. Nonetheless, the images in new Fig. 4 (at 1-h incubation) has been improved and may be published as images after a long-term (1 h) incubation (although I still think that showing the structures after short-term incubation and a FRAP analysis would strengthen the conclusion).

In our conditions, we have observed p62 droplets with different time lengths (10, 20, 30, 60 min), and consistently saw p62 assemblies had similar shapes, as shown in Figure 4. This suggests that p62 may quickly form gel-like structures, as Reviewer 3 pointed previously, and this may reflect the fact that DAXX promotes p62 phase condensation.

The authors have now shown the effects of ATG gene knockdown on p62 accumulation and LC3-II formation as a figure for reviewers. However, the differences are not large, and LC3-I does not accumulate. I suggest that this data should be published in supplemental figures so that readers can judge how much the knockdown of these proteins is effective.

We agree on the point that these data should be published for readers' judgement, and the figures are now moved as Supplementary Figure 7c.

In these experiments, we did not observe LC3-I accumulation. In HeLa cells, LC3-II is the prominent form of LC3, and the LC3 antibody (CST, #12741) used here predominantly picks up LC3-II.

We are thankful to the Reviewer for his/her time and efforts on the manuscript.